# Noisy Recurrent Neural Networks

**Soon Hoe Lim**
Nordita, KTH Royal Institute of Technology
and Stockholm University
soon.hoe.lim@su.se

**N. Benjamin Erichson**
University of Pittsburgh
School of Engineering
erichson@pitt.edu

**Liam Hodgkinson**
ICSI and Department of Statistics
UC Berkeley
liam.hodgkinson@berkeley.edu

**Michael W. Mahoney**
ICSI and Department of Statistics
UC Berkeley
mmahoney@stat.berkeley.edu

## Abstract

We provide a general framework for studying recurrent neural networks (RNNs) trained by injecting noise into hidden states. Specifically, we consider RNNs that can be viewed as discretizations of stochastic differential equations driven by input data. This framework allows us to study the implicit regularization effect of general noise injection schemes by deriving an approximate explicit regularizer in the small noise regime. We find that, under reasonable assumptions, this implicit regularization promotes flatter minima; it biases towards models with more stable dynamics; and, in classification tasks, it favors models with larger classification margin. Sufficient conditions for global stability are obtained, highlighting the phenomenon of stochastic stabilization, where noise injection can improve stability during training. Our theory is supported by empirical results which demonstrate that the RNNs have improved robustness with respect to various input perturbations.

## 1 Introduction

Viewing recurrent neural networks (RNNs) as discretizations of ordinary differential equations (ODEs) driven by input data has recently gained attention [7, 27, 16, 49]. The "formulate in continuous time, and then discretize" approach [38] motivates novel architecture designs before experimentation, and it provides a useful interpretation as a dynamical system. This, in turn, has led to gains in reliability and robustness to data perturbations.

Recent efforts have shown how adding noise can also improve stability during training, and consequently improve robustness [35]. In this work, we consider discretizations of the corresponding stochastic differential equations (SDEs) obtained from ODE formulations of RNNs through the addition of a diffusion (noise) term. We refer to these as *Noisy RNNs* (NRNNs). By dropping the noisy elements at inference time, NRNNs become a stochastic learning strategy which, as we shall prove, has a number of important benefits. In particular, stochastic learning strategies (including dropout) are often used as natural regularizers, favoring solutions in regions of the loss landscape with desirable properties (often improved generalization and/or robustness). This mechanism is commonly referred to as *implicit regularization* [40, 39, 50], differing from *explicit regularization* where the loss is explicitly modified. For neural network (NN) models, implicit regularization towards wider minima is conjectured to be a prominent ingredient in the success of stochastic optimization [67, 28]. Indeed, implicit regularization has been linked to increases in classification margins [47], which can lead to improved generalization performance [51]. A common approach to identify and study implicit regularization is to approximate the implicit regularization by an appropriate explicit regularizer

35th Conference on Neural Information Processing Systems (NeurIPS 2021).

[40, 39, 1, 6, 21]. Doing so, we will see that NRNNs favor wide minima (like SGD); more stable dynamics; and classifiers with a large classification margin, keeping generalization error small.

SDEs have also seen recent appearances in *neural SDEs* [59, 24], stochastic generalizations of *neural ODEs* [9] which can be seen as an analogue of NRNNs for non-sequential data, with a similar relationship to NRNNs as feedforward NNs do to RNNs. They have been shown to be robust in practice [35]. Analogously, we shall show that the NRNN framework leads to more reliable and robust RNN classifiers, whose promise is demonstrated by experiments on benchmark data sets.

**Contributions.** For the class of NRNNs (formulated first as a continuous-time model, which is then discretized), the following are our main contributions:

- we identify the form of the implicit regularization for NRNNs through a corresponding (data-dependent) explicit regularizer in the small noise regime (see Theorem 1);

- we focus on its effect in classification tasks, providing bounds for the classification margin for the deterministic RNN classifiers (see Theorem 2); in particular, Theorem 2 reveals that *stable RNN dynamics can lead to large classification margin*;

- we show that noise injection can also lead to improved stability (see Theorem 3) via a Lyapunov stability analysis of continuous-time NRNNs;

- we demonstrate via empirical experiments on benchmark data sets that NRNN classifiers are more robust to data perturbations when compared to other recurrent models, while retaining state-of-the-art performance for clean data. Research code is provided here: https://github.com/erichson/NoisyRNN.

**Notation.** We use $\|v\| := \|v\|_2$ to denote the Euclidean norm of the vector $v$, and $\|A\|_2$ and $\|A\|_F$ to denote the spectral norm and Frobenius norm of the matrix $A$, respectively. The $i$th element of a vector $v$ is denoted by $v^i$ or $[v]^i$, and the $(i,j)$-entry of a matrix $A$ by $A^{ij}$ or $[A]^{ij}$. For a vector $v = (v^1, \ldots, v^d)$, $\text{diag}(v)$ denotes the diagonalization of $v$ with $\text{diag}(v)^{ii} = v^i$. $I$ denotes the identity matrix (with dimension clear from context), while superscript $T$ denotes transposition. For a matrix $M$, $M^{\text{sym}} = (M + M^T)/2$ denotes its symmetric part, $\lambda_{\min}(M)$ and $\lambda_{\max}(M)$ denote its minimum and maximum eigenvalue respectively, $\sigma_{\max}(M)$ denotes its maximum singular value, and $Tr(M)$ denotes its trace. For a function $f : \mathbb{R}^n \to \mathbb{R}^m$ such that each of its first-order partial derivatives (with respect to $x$) exist, $\frac{\partial f}{\partial x} \in \mathbb{R}^{m \times n}$ is the Jacobian matrix of $f$. For a scalar-valued function $g : \mathbb{R}^n \to \mathbb{R}$, $\nabla_h g$ is the gradient of $g$ with respect to the variable $h \in \mathbb{R}^n$ and $H_h g$ is the Hessian of $g$ with respect to $h$.

## 2    Related Work

**Dynamical Systems and Machine Learning.** There are various interesting connections between machine learning and dynamical systems. Formulating machine learning in the framework of continuous-time dynamical systems was recently popularized by [62]. Subsequent efforts focus on constructing learning models by approximating continuous-time dynamical systems [9, 29, 48] and studying them using tools from numerical analysis [36, 64, 69, 68]. On the other hand, dynamical systems theory provides useful theoretical tools for analyzing NNs, including RNNs [60, 15, 34, 7, 16], and useful principles for designing NNs [23, 54]. Other examples of dynamical systems inspired models include the learning of invariant quantities via their Hamiltonian or Lagrangian representations [37, 22, 10, 71, 58]. Another class of models is inspired by Koopman theory, yielding models where the evolution operator is linear [56, 43, 17, 45, 33, 4, 3, 13].

**Stochastic Training and Regularization Strategies.** Regularization techniques such as noise injection and dropout can help to prevent overfitting in NNs. Following the classical work [5] that studies regularizing effects of noise injection on data, several work studies the effects of noise injection into different parts of networks for various architectures [25, 44, 35, 54, 2, 26, 66, 61]. In particular, recently [6] studies the regularizing effect of isotropic Gaussian noise injection into the layers of feedforward networks. For RNNs, [12] shows that noise additions on the hidden states outperform Bernoulli dropout in terms of performance and bias, whereas [18] introduces a variant of stochastic RNNs for generative modeling of sequential data. Some specific formulations of RNNs as SDEs were

also considered in Chapter 10 of [41]. Implicit regularization has also been studied more generally than stochastic gradient based training of NNs [40, 39, 19, 11].

# 3 Noisy Recurrent Neural Networks

We formulate continuous-time recurrent neural networks (CT-RNNs) at full generality as a system of input-driven ODEs: for a terminal time $T > 0$ and an input signal $x = (x_t)_{t \in [0,T]} \in C([0,T]; \mathbb{R}^{d_x})$, the output $y_t \in \mathbb{R}^{d_y}$, for $t \in [0,T]$, is a linear map of hidden states $h_t \in \mathbb{R}^{d_h}$ satisfying

$$\mathrm{d}h_t = f(h_t, x_t)\mathrm{d}t, \qquad y_t = V h_t, \qquad (1)$$

where $V \in \mathbb{R}^{d_y \times d_h}$, and $f : \mathbb{R}^{d_h} \times \mathbb{R}^{d_x} \to \mathbb{R}^{d_h}$ is typically Lipschitz continuous, guaranteeing existence and uniqueness of solutions to (1).

A natural stochastic variant of CT-RNNs arises by replacing the ODE in (1) by an Itô SDE, that is,

$$\mathrm{d}h_t = f(h_t, x_t)\mathrm{d}t + \sigma(h_t, x_t)\mathrm{d}B_t, \qquad y_t = V h_t, \qquad (2)$$

where $\sigma : \mathbb{R}^{d_h} \times \mathbb{R}^{d_x} \to \mathbb{R}^{d_h \times r}$ and $(B_t)_{t \geq 0}$ is an $r$-dimensional Brownian motion. The functions $f, \sigma$ are referred to as the *drift* and *diffusion* coefficients, respectively. Intuitively, (2) amounts to a noisy perturbation of the corresponding deterministic CT-RNN (1). At full generality, we refer to the system (2) as a *continuous-time Noisy RNN* (CT-NRNN). To guarantee the existence of a unique solution to (2), in the sequel, we assume that $\{f(\cdot, x_t)\}_{t \in [0,T]}$ and $\{\sigma(\cdot, x_t)\}_{t \in [0,T]}$ are uniformly Lipschitz continuous, and $t \mapsto f(h, x_t)$, $t \mapsto \sigma(h, x_t)$ are bounded in $t \in [0,T]$ for each fixed $h \in \mathbb{R}^{d_h}$. For further details, see Section B in Supplementary Material (**SM**).

While much of our theoretical analysis will focus on this general formulation of CT-NRNNs, our empirical and stability analyses focus on the choice of drift function

$$f(h, x) = Ah + a(Wh + Ux + b), \qquad (3)$$

where $a : \mathbb{R} \to \mathbb{R}$ is a Lipschitz continuous scalar activation function extended to act on vectors pointwise, $A, W \in \mathbb{R}^{d_h \times d_h}$, $U \in \mathbb{R}^{d_h \times d_x}$ and $b \in \mathbb{R}^{d_h}$. Typical examples of activation functions include $a(x) = \tanh(x)$. The matrices $A, W, U, V, b$ are all assumed to be trainable parameters. This particular choice of drift dates back to the early Cohen-Grossberg formulation of CT-RNNs, and was recently reconsidered in [16].

## 3.1 Noise Injections as Stochastic Learning Strategies

While precise choices of drift functions $f$ are the subject of existing deterministic RNN theory, good choices of the diffusion coefficient $\sigma$ are less clear. Here, we shall consider a parametric class of diffusion coefficients given by:

$$\sigma(h, x) \equiv \epsilon(\sigma_1 I + \sigma_2 \mathrm{diag}(f(h, x))), \qquad (4)$$

where the noise level $\epsilon > 0$ is small, and $\sigma_1 \geq 0$ and $\sigma_2 \geq 0$ are tunable parameters describing the relative strength of additive noise and a multiplicative noise respectively.

While the stochastic component is an important part of the model, one can set $\epsilon \equiv 0$ at inference time. In doing so, noise injections in NRNNs may be viewed as a learning strategy. A similar stance is considered in [35] for treating neural SDEs. From this point of view, we may relate noise injections generally to regularization mechanisms considered in previous works. For example, additive noise injection was studied in the context of feedforward NNs in [6], in which case a Gaussian noise is injected to the activation function at each layer of the NN. Furthermore, multiplicative noise injections includes stochastic depth and dropout strategies as special cases [36, 35]. By taking a Gaussian approximation to Bernoulli noise and taking a continuous-time limit, NNs with stochastic dropout can be weakly approximated by an SDE with appropriate multiplicative noise, see [36]. All of these works highlight various advantages of noise injection for training NNs.

## 3.2 Numerical Discretizations

As in the deterministic case, exact simulation of the SDE in (2) is infeasible in practice, and so one must specify a numerical integration scheme. We will focus on the explicit Euler-Maruyama (E-M) integrators [30], which are the stochastic analogues of Euler-type integration schemes for ODEs.

Let $0 := t_0 < t_1 < \cdots < t_M := T$ be a partition of the interval $[0, T]$. Denote $\delta_m := t_{m+1} - t_m$ for each $m = 0, 1, \ldots, M-1$, and $\delta := (\delta_m)$. The E-M scheme provides a family (parametrized by $\delta$) of approximations to the solution of the SDE in (2):

$$h_{m+1}^\delta = h_m^\delta + f(h_m^\delta, \hat{x}_m)\delta_m + \sigma(h_m^\delta, \hat{x}_m)\sqrt{\delta_m}\xi_m, \tag{5}$$

for $m = 0, 1, \ldots, M-1$, where $(\hat{x}_m)_{m=0,\ldots,M-1}$ is a given sequential data, the $\xi_m \sim \mathcal{N}(0, I)$ are independent $r$-dimensional standard normal random vectors, and $h_0^\delta = h_0$. As $\Delta := \max_m \delta_m \to 0$, the family of approximations $(h_m^\delta)$ converges strongly to the Itô process $(h_t)$ satisfying (2) (at rate $\mathcal{O}(\sqrt{\Delta})$ when the step sizes are uniform; see Theorem 10.2.2 in [30]). See Section C in **SM** for details on the general case.

## 4 Implicit Regularization

To highlight the advantages of NRNNs over their deterministic counterpart, we show that, under reasonable assumptions, NRNNs exhibit a natural form of *implicit regularization*. By this, we mean regularization imposed implicitly by the stochastic learning strategy, without explicitly modifying the loss, but that, e.g., may promote flatter minima. Our goal is achieved by deriving an appropriate explicit regularizer through a perturbation analysis in the small noise regime. This becomes useful when considering NRNNs as a learning strategy, since we can precisely determine the effect of the noise injection as a regularization mechanism.

The study for discrete-time NRNNs is of practical interest and is our focus here. Nevertheless, analogous results for continuous-time NRNNs are also valuable for exploring other discretization schemes. For this reason, we also study the continuous-time case in Section E in **SM**. Our analysis covers general NRNNs, not necessarily those with the drift term (3) and diffusion term (4), that satisfy the following assumption, which is typically reasonable in practice. We remark that a ReLU activation will violate the assumption. However, RNNs with ReLU activation are less widely used in practice. Without careful initialization [31, 57], they typically suffer more from exploding gradient problems compared to those with bounded activation functions such as $\tanh$.

**Assumption A.** The drift $f$ and diffusion coefficient $\sigma$ of the SDE in (2) satisfy the following:

  (i) for all $t \in [0, T]$ and $x \in \mathbb{R}^{d_x}$, $h \mapsto f(h, x)$ and $h \mapsto \sigma^{ij}(h, x)$ have Lipschitz continuous partial derivatives in each coordinate up to order three (inclusive);

  (ii) for any $h \in \mathbb{R}^{d_h}$, $t \mapsto f(h, x_t)$ and $t \mapsto \sigma(h, x_t)$ are bounded and Borel measurable on $[0, T]$.

We consider a rescaling of the noise $\sigma \mapsto \epsilon\sigma$ in (2), where $\epsilon > 0$ is assumed to be a small parameter, in line with our noise injection strategies in Subsection 3.1.

In the sequel, we let $\bar{h}_m^\delta$ denote the hidden states of the corresponding deterministic RNN model, satisfying

$$\bar{h}_{m+1}^\delta = \bar{h}_m^\delta + \delta_m f(\bar{h}_m^\delta, \hat{x}_m), \quad m = 0, 1, \ldots, M-1, \tag{6}$$

with $\bar{h}_0^\delta = h_0$. Let $\Delta := \max_{m \in \{0,\ldots,M-1\}} \delta_m$, and denote the state-to-state Jacobians by

$$\hat{J}_m = I + \delta_m \frac{\partial f}{\partial h}(\bar{h}_m^\delta, \hat{x}_m). \tag{7}$$

For $m, k = 0, \ldots, M-1$, also let

$$\hat{\Phi}_{m,k} = \hat{J}_m \hat{J}_{m-1} \cdots \hat{J}_k, \tag{8}$$

where the empty product is assumed to be the identity. Note that the $\hat{\Phi}_{m,k}$ are products of the state-to-state Jacobian matrices, important for analyzing signal propagation in RNNs [8]. For the sake of brevity, we denote $f_m = f(\bar{h}_m^\delta, \hat{x}_m)$ and $\sigma_m = \sigma(\bar{h}_m^\delta, \hat{x}_m)$ for $m = 0, 1, \ldots, M$.

The following result, which is our first main result, relates the loss function, averaged over realizations of the injected noise, used for training NRNN to that for training deterministic RNN in the small noise regime.

**Theorem 1** (Implicit regularization induced by noise injection)**.** *Under Assumption A,*

$$\mathbb{E}\ell(h_M^\delta) = \ell(\bar{h}_M^\delta) + \frac{\epsilon^2}{2}[\hat{Q}(\bar{h}^\delta) + \hat{R}(\bar{h}^\delta)] + \mathcal{O}(\epsilon^3), \tag{9}$$

as $\epsilon \to 0$, where the terms $\hat{Q}$ and $\hat{R}$ are given by

$$\hat{Q}(\bar{h}^\delta) = \nabla l(\bar{h}_M^\delta)^T \sum_{k=1}^{M} \delta_{k-1} \hat{\Phi}_{M-1,k} \sum_{m=1}^{M-1} \delta_{m-1} \boldsymbol{v}_m, \tag{10}$$

$$\hat{R}(\bar{h}^\delta) = \sum_{m=1}^{M} \delta_{m-1} \mathrm{tr}(\sigma_{m-1}^T \hat{\Phi}_{M-1,m}^T H_{\bar{h}^\delta} l \; \hat{\Phi}_{M-1,m} \sigma_{m-1}), \tag{11}$$

with $\boldsymbol{v}_m$ a vector with the pth component:

$$[v_m]^p = \mathrm{tr}(\sigma_{m-1}^T \hat{\Phi}_{M-2,m}^T H_{\bar{h}^\delta} [f_M]^p \hat{\Phi}_{M-2,m} \sigma_{m-1}), \tag{12}$$

for $p = 1, \ldots, d_h$. Moreover,

$$|\hat{Q}(\bar{h}^\delta)| \le C_Q \Delta^2, \;\; |\hat{R}(\bar{h}^\delta)| \le C_R \Delta, \tag{13}$$

for $C_Q, C_R > 0$ independent of $\Delta$.

If the loss is convex, then $\hat{R}$ is non-negative, but $\hat{Q}$ needs not be. However, $\hat{Q}$ can be made negligible relative to $\hat{R}$ provided that $\Delta$ is taken sufficiently small. This also ensures that the E-M approximations are accurate.

To summarize, Theorem 1 implies that the injection of noise into the hidden states of deterministic RNN is, on average, approximately equivalent to a regularized objective functional. Moreover, the explicit regularizer is solely determined by the discrete-time flow generated by the Jacobians $\frac{\partial f_m}{\partial h}(\bar{h}_m^\delta)$, the diffusion coefficients $\sigma_n$, and the Hessian of the loss function, all evaluated along the dynamics of the deterministic RNN. We can therefore expect that the use of NRNNs as a regularization mechanism should reduce the state-to-state Jacobians and Hessian of the loss function according to the noise level $\epsilon$. Indeed, NRNNs exhibit a smoother Hessian landscape than that of the deterministic counterpart (see Figure 3 in **SM**).

The Hessian of the loss function commonly appears in implicit regularization analyses, and suggests a preference towards wider minima in the loss landscape. Commonly considered a positive attribute [28], this, in turn, suggests a degree of robustness in the loss to perturbations in the hidden states [65]. More interesting, however, is the appearance of the Jacobians, which is indicative of a preference towards slower, more stable dynamics. Both of these attributes suggest NRNNs could exhibit a strong tendency towards models which are less sensitive to input perturbations. Overall, we can see that the use of NRNNs as a regularization mechanism reduces the state-to-state Jacobians and Hessian of the loss function according to the noise level.

## 5 Implications in Classification Tasks

Our focus now turns to an investigation of the benefits of NRNNs over their deterministic counterparts for classification tasks. From Theorem 1, it is clear that adding noise to deterministic RNN implicitly regularizes the state-to-state Jacobians. Here, we show that doing so also enhances an implicit tendency towards classifiers with large classification margin. Our analysis here covers general deterministic RNNs, although we also apply our results to obtain explicit expressions for Lipschitz RNNs.

Let $\mathcal{S}_N$ denote a set of training samples $s_n := (\boldsymbol{x}_n, y_n)$ for $n = 1, \ldots, N$, where each input sequence $\boldsymbol{x}_n = (x_{n,0}, x_{n,1}, \ldots, x_{n,M-1}) \in \mathcal{X} \subset \mathbb{R}^{d_x M}$ has a corresponding class label $y_n \in \mathcal{Y} = \{1, \ldots, d_y\}$. Following the statistical learning framework, these samples are assumed to be independently drawn from an underlying probability distribution $\mu$ on the sample space $\mathcal{S} = \mathcal{X} \times \mathcal{Y}$. An RNN-based classifier $g^\delta(\boldsymbol{x})$ is constructed in the usual way by taking

$$g^\delta(\boldsymbol{x}) = \mathrm{argmax}_{i=1,\ldots,d_y} p^i(V\bar{h}_M^\delta[\boldsymbol{x}]), \tag{14}$$

where $p^i(x) = e^{x^i}/\sum_j e^{x^j}$ is the softmax function. Letting $\ell$ denoting the cross-entropy loss, such a classifier is trained from $\mathcal{S}_N$ by minimizing the empirical risk (training error), $\mathcal{R}_N(g^\delta) := \frac{1}{N}\sum_{n=1}^{N} \ell(g^\delta(\boldsymbol{x}_n), y_n)$, as a proxy for the true (population) risk (test error), $\mathcal{R}(g^\delta) =$

$\mathbb{E}_{(\boldsymbol{x},y)\sim\mu}\ell(g^\delta(\boldsymbol{x}),y)$, with $(\boldsymbol{x},y)\in\mathcal{S}$. The measure used to quantify the prediction quality is the generalization error (or estimation error), which is the difference between the empirical risk of the classifier on the training set and the true risk: $\mathrm{GE}(g^\delta):=|\mathcal{R}(g^\delta)-\mathcal{R}_N(g^\delta)|$.

The classifier is a function of the output of the deterministic RNN, which is an Euler discretization of the ODE (1) with step sizes $\delta=(\delta_m)$. In particular, for the Lipschitz RNN,

$$\hat{\Phi}_{m,k}=\hat{J}_m\hat{J}_{m-1}\cdots\hat{J}_k, \tag{15}$$

where $\hat{J}_l=I+\delta_l(A+D_lW)$, with $D_l^{ij}=a'([W\bar{h}_l^\delta+U\hat{x}_l+b]^i)e_{ij}$.

In the following, we let $\mathrm{conv}(\mathcal{X})$ denote the convex hull of $\mathcal{X}$. We let $\hat{\boldsymbol{x}}_{0:m}:=(\hat{x}_0,\ldots,\hat{x}_m)$ so that $\hat{\boldsymbol{x}}=\hat{\boldsymbol{x}}_{0:M-1}$, and use the notation $f[\boldsymbol{x}]$ to indicate the dependence of the function $f$ on the vector $\boldsymbol{x}$. Our result will depend on two characterizations of a training sample $s_i=(\boldsymbol{x}_i,y_i)$.

**Definition 1** (Classification Margin). The classification margin of a training sample $s_i=(\boldsymbol{x}_i,y_i)$ measured by the Euclidean metric $d$ is defined as the radius of the largest $d$-metric ball in $\mathcal{X}$ centered at $\boldsymbol{x}_i$ that is contained in the decision region associated with the class label $y_i$, i.e., it is: $\gamma^d(s_i)=\sup\{a:d(\boldsymbol{x}_i,\boldsymbol{x})\le a\Rightarrow g^\delta(\boldsymbol{x})=y_i\ \ \forall\boldsymbol{x}\}$.

Intuitively, a larger classification margin allows a classifier to associate a larger region centered on a point $\boldsymbol{x}_i$ in the input space to the same class. This makes the classifier less sensitive to input perturbations, and a perturbation of $\boldsymbol{x}_i$ is still likely to fall within this region, keeping the classifier prediction. In this sense, the classifier becomes more robust. In our case, the networks are trained by a loss (cross-entropy) that promotes separation of different classes in the network output. This, in turn, maximizes a certain notion of score of each training sample.

**Definition 2** (Score). For a training sample $s_i=(\boldsymbol{x}_i,y_i)$, we define its score as $o(s_i)=\min_{j\ne y_i}\sqrt{2}(e_{y_i}-e_j)^TS^\delta[\boldsymbol{x}_i]\ge 0$, where $e_i\in\mathbb{R}^{d_y}$ is the Kronecker delta vector with $e_i^i=1$ and $e_i^j=0$ for $i\ne j$, $S^\delta[\boldsymbol{x}_i]:=p(V\bar{h}_M^\delta[\boldsymbol{x}_i])$ with $\bar{h}_M^\delta[\boldsymbol{x}_i]$ denoting the hidden state of the RNN, driven by the input sequence $\boldsymbol{x}_i$, at terminal index $M$.

Recall that the classifier $g^\delta(\boldsymbol{x})=\arg\max_{i\in 1,\ldots,d_y}[S^\delta]^i[\boldsymbol{x}]$, and the decision boundary between class $i$ and class $j$ in the feature space is given by the hyperplane $\{z=S^\delta:z^i=z^j\}$. A positive score implies that at the network output, classes are separated by a margin that corresponds to the score. However, a large score may not imply a large classification margin.

Following the approach of [52, 63], we obtain the second main result, providing bounds for classification margin for the deterministic RNN classifiers $g^\delta$. We also provide a generalization bound in terms of the classification margin under additional assumptions (see Theorem 11 in **SM**).

**Theorem 2.** *Suppose that Assumption A holds. Assume that the score $o(s_i)>0$ and*

$$\gamma(s_i):=\frac{o(s_i)}{C\sum_{m=0}^{M-1}\delta_m\sup_{\hat{\boldsymbol{x}}\in\mathrm{conv}(\mathcal{X})}\|\hat{\Phi}_{M,m+1}[\hat{\boldsymbol{x}}]\|_2}>0, \tag{16}$$

*where $C=\|V\|_2\left(\max_{m=0,1,\ldots,M-1}\left\|\frac{\partial f(\bar{h}_m^\delta,\hat{x}_m)}{\partial\hat{x}_m}\right\|_2\right)>0$ is independent of $s_i$ (in particular, $C=\|V\|_2[\max_{m=0,\ldots,M-1}\|D_mU\|_2]$ for Lipschitz RNNs), the $\hat{\Phi}_{m,k}$ are defined in (15) and the $\delta_m$ are the step sizes. Then, the classification margin for the training sample $s_i$:*

$$\gamma^d(s_i)\ge\gamma(s_i). \tag{17}$$

Now, recalling from Section 4, up to $\mathcal{O}(\epsilon^2)$ and under the assumption that $\hat{Q}$ vanishes, the loss minimized by the NRNN classifer is, on average, $\ell(\bar{h}_M^\delta)+\epsilon^2\hat{R}(\bar{h}^\delta)$, as $\epsilon\to 0$, with regularizer

$$\hat{R}(\bar{h}^\delta)=\frac{1}{2}\sum_{m=1}^M\delta_{m-1}\|\hat{M}_{M-1}\hat{\Phi}_{M-1,m}\sigma_{m-1}\|_F^2, \tag{18}$$

where $\hat{M}_M^T\hat{M}_M:=H_{\bar{h}_M^\delta}l$ is the Cholesky decomposition of the Hessian matrix of the convex cross-entropy loss. The appearance of the state-to-state Jacobians in $\Phi_{m,k}$ in both the regularizer (18) and the lower bound (16) suggests that noise injection implicitly aids generalization performance. More precisely, in the small noise regime and on average, NRNNs promote classifiers with large

classification margin, an attribute linked to both improved robustness and generalization [63]. In this sense, training with NRNN classifiers is a stochastic strategy to improve generalization over deterministic RNN classifiers, particularly in learning tasks where the given data is corrupted (c.f. the caveats pointed out in [53]).

Theorem 2 implies that the lower bound for the classification margin is determined by the spectrum of the $\hat{\Phi}_{M-1,m}$. To make the lower bound large, keeping $\delta_m$ and $M$ fixed, the spectral norm of the $\hat{\Phi}_{M-1,m}$ should be made small. Doing so improves stability of the RNN, but may also lead to vanishing gradients, hindering capacity of the model to learn. To maximize the lower bound while avoiding the vanishing gradient problem, one should tune the numerical step sizes $\delta_m$ and noise level $\epsilon$ in NRNN appropriately. RNN architectures for the drift which help to ensure moderate Jacobians (e.g., $\|\hat{\Phi}_{M-1,m}\|_2 \approx 1$ for all $m$ [8]) also remain valuable in this respect.

# 6 Stability and Noise-Induced Stabilization

Here we obtain sufficient conditions to guarantee stochastic stability of CT-NRNNs. This will also provide another lens to highlight the potential of NRNNs for improved robustness. A dynamical system is considered *stable* if trajectories which are close to each other initially remain close at subsequent times. As observed in [46, 42, 7], stability plays an essential role in the study of RNNs to avoid the *exploding gradient problem*, a property of unstable systems where the gradient increases in magnitude with the depth. While gradient clipping during training can somewhat alleviate this issue, better performance and robustness is achieved by enforcing stability in the model itself.

Our stability analysis will focus on establishing *almost sure exponential stability* (for other notions of stability, see **SM**) for CT-NRNNs with the drift function (3). To preface the definition, consider initializing the SDE at two different random variables $h_0$ and $h'_0 := h_0 + \epsilon_0$, where $\epsilon_0 \in \mathbb{R}^{d_h}$ is a constant non-random perturbation with $\|\epsilon_0\| \leq \delta$. The resulting hidden states, $h_t$ and $h'_t$, are set to satisfy (2) with the same Brownian motion $B_t$, starting from their initial values $h_0$ and $h'_0$, respectively. The evolution of $\epsilon_t = h'_t - h_t$ satisfies

$$d\epsilon_t = A\epsilon_t dt + \Delta a_t(\epsilon_t)dt + \Delta \sigma_t(\epsilon_t)dB_t, \tag{19}$$

where $\Delta a_t(\epsilon_t) = a(Wh'_t + Ux_t + b) - a(Wh_t + Ux_t + b)$ and $\Delta \sigma_t(\epsilon_t) = \sigma(h_t + \epsilon_t, x_t) - \sigma(h_t, x_t)$. Since $\Delta a_t(0) = 0$, $\Delta \sigma_t(0) = 0$ for all $t \in [0, T]$, $\epsilon_t = 0$ admits a trivial *equilibrium* for (19). Our objective is to analyze the stability of the solution $\epsilon_t = 0$, that is, to see how the final state $\epsilon_T$ (and hence the output of the RNN) changes for an arbitrarily small initial perturbation $\epsilon_0 \neq 0$. To this end, we consider an extension of the Lyapunov exponent to SDEs at the level of sample path [41].

**Definition 3** (Almost sure global exponential stability)**.** The sample (or pathwise) Lyapunov exponent of the trivial solution of (19) is $\Lambda = \limsup_{t\to\infty} t^{-1} \log \|\epsilon_t\|$. The trivial solution $\epsilon_t = 0$ is *almost surely globally exponentially stable* if $\Lambda$ is almost surely negative for all $\epsilon_0 \in \mathbb{R}^{d_h}$.

For the sample Lyapunov exponent $\Lambda(\omega)$, there is a constant $C > 0$ and a random variable $0 \leq \tau(\omega) < \infty$ such that for all $t > \tau(\omega)$, $\|\epsilon_t\| = \|h'_t - h_t\| \leq Ce^{\Lambda t}$ almost surely. Therefore, almost sure exponential stability implies that almost all sample paths of (19) will tend to the equilibrium solution $\epsilon = 0$ exponentially fast. With this definition in tow, we obtain the following stability result.

**Theorem 3.** *Assume that $a$ is monotone non-decreasing, and $\sigma_1\|\epsilon\| \leq \|\Delta\sigma_t(\epsilon)\|_F \leq \sigma_2\|\epsilon\|$ for all nonzero $\epsilon \in \mathbb{R}^{d_h}$, $t \in [0, T]$. Then for any $\epsilon_0 \in \mathbb{R}^{d_h}$, with probability one,*

$$\phi + \lambda_{\min}(A^{\mathrm{sym}}) \leq \Lambda \leq \psi + L_a\sigma_{\max}(W) + \lambda_{\max}(A^{\mathrm{sym}}), \tag{20}$$

*with $\phi = -\sigma_2^2 + \frac{\sigma_1^2}{2}$ and $\psi = -\sigma_1^2 + \frac{\sigma_2^2}{2}$, where $L_a$ is the Lipschitz constant of $a$.*

In the special case without noise ($\sigma_1 = \sigma_2 = 0$), we recover case (a) of Theorem 1 in [16]: when $A^{\mathrm{sym}}$ is negative definite and $\sigma_{\min}(A^{\mathrm{sym}}) > L_a\sigma_{\max}(W)$, Theorem 3 implies that (2) is exponentially stable. Most strikingly, and similar to [35], Theorem 3 implies that even if the deterministic CT-RNN is not exponentially stable, it can be stabilized through a stochastic perturbation. Consequently, injecting noise appropriately can improve training performance.

# 7 Empirical Results

The evaluation of robustness of neural networks (RNNs in particular) is an often neglected yet crucial aspect. In this section, we investigate the robustness of NRNNs and compare their performance to other recently introduced state-of-the-art models on both clean and corrupted data. We refer to Section G in **SM** for further details of our experiments.

Here, we study the sensitivity of different RNN models with respect to a sequence of perturbed inputs during inference time. We consider different types of perturbations: (a) white noise; (b) multiplicative white noise; (c) salt and pepper; and (d) adversarial perturbations. To be more concrete, let $x$ be a sequence. The perturbations in consideration are as follows.

- *Additive white noise perturbations* are constructed as $\tilde{x} = x + \Delta x$, where the additive noise is drawn from a Gaussian distribution $\Delta x \sim \mathcal{N}(0, \sigma)$. This perturbation strategy emulates measurement errors that can result from data acquisition with poor sensors (where $\sigma$ can be used to vary the strength of these errors). *Multiplicative white noise perturbations* are constructed as $\tilde{x} = x \cdot \Delta x$, where the additive noise is drawn from a Gaussian distribution $\Delta x \sim \mathcal{N}(1, \sigma_M)$.

- *Salt and pepper perturbations* emulate defective pixels that result from converting analog signals to digital signals. The noise model takes the form $\mathbb{P}(\tilde{X} = X) = 1 - \alpha$, and $\mathbb{P}(\tilde{X} = \max) = \mathbb{P}(\tilde{X} = \min) = \alpha/2$, where $\tilde{X}(i,j)$ denotes the corrupted image and $\min$ and $\max$ denote to the minimum and maximum pixel values. The parameter $\alpha$ controls the proportion of defective pixels.

- *Adversarial perturbations* are "worst-case" non-random perturbations maximizing the loss $\ell(g^\delta(X + \Delta X), y)$ subject to the constraint that the norm of the perturbation $\|\Delta X\| \le r$. We consider the fast gradient sign method for constructing these perturbations [55].

We consider in addition to the NRNN three other RNNs derived from continuous-time models, including the Lipschitz RNN [16] (the deterministic counterpart to our NRNN), the coupled oscillatory RNN (coRNN) [49] and the antisymmetric RNN [7]. We also consider the exponential RNN [32], a discrete-time model that uses orthogonal recurrent weights. We train each model with the prescribed tuning parameters for the ordered (see Sec. 7.1) and permuted (see **SM**) MNIST task. For the Electrocardiogram (ECG) classification task we performed a non-exhaustive hyper-tuning parameter search. For comparison, we train all models with hidden-to-hidden weight matrices of dimension $d_h = 128$. We average the classification performance over ten different seed values.

## 7.1 Ordered Pixel-by-Pixel MNIST Classification

First, we consider the ordered pixel-by-pixel MNIST classification task [31]. This task sequentially presents 784 pixels to the model and uses the final hidden state to predict the class membership probability of the input image. In the **SM** we present additional results for the situation when instead of an ordered sequence a fixed random permutation of the input sequence is presented to the model.

Table 1 shows the average test accuracy (evaluated for models that are trained with 10 different seed values) for the ordered task. Here we present results for white noise and salt and pepper (S&P) perturbations. While the Lipschitz RNN performs best on clean input sequences, the NRNNs show an improved resilience to input perturbations. Here, we consider two different configuration for the NRNN. In both cases, we set the multiplicative noise level to $0.02$, whereas we consider the additive noise levels $0.02$ and $0.05$. We chose these configurations as they appear to provide a good trade-off between accuracy and robustness. Note that the predictive accuracy on clean inputs starts to drop when the noise level becomes too large.

Table 1: Robustness w.r.t. white noise ($\sigma$) and S&P ($\alpha$) perturbations on the ordered MNIST task.

| Name | clean | $\sigma = 0.1$ | $\sigma = 0.2$ | $\sigma = 0.3$ | $\alpha = 0.03$ | $\alpha = 0.05$ | $\alpha = 0.1$ |
|---|---|---|---|---|---|---|---|
| Antisymmetric RNN [7] | 97.5% | 45.7% | 22.3% | 17.0% | 77.1% | 63.9% | 42.6% |
| CoRNN [49] | 99.1% | 96.6% | 61.9% | 32.1% | 95.6% | 88.1% | 58.9% |
| Exponential RNN [32] | 96.7% | 86.7% | 58.1% | 33.3% | 83.6% | 70.7% | 43.4% |
| Lipschitz RNN [16] | **99.2**% | 98.4% | 78.9% | 47.1% | 97.6% | 93.4% | 73.5% |
| NRNN (mult./add. noise: 0.02/0.02) | 99.1% | **98.9**% | 88.4% | 62.9% | 98.3% | 95.6% | 78.7% |
| NRNN (mult./add. noise: 0.02/0.05) | 99.1% | **98.9**% | **92.2**% | **73.5**% | **98.5**% | **97.1**% | **85.5**% |

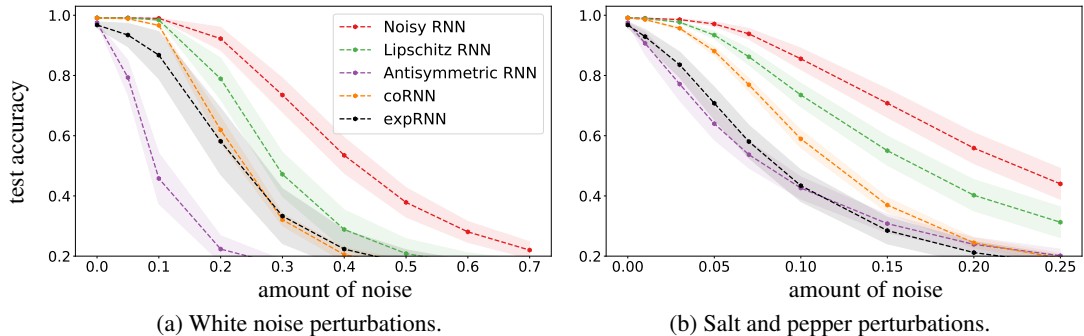

| (a) White noise perturbations. | (b) Salt and pepper perturbations. |

Figure 1: Test accuracy for the ordered MNIST task as function of the strength of input perturbations.

Table 2 shows the average test accuracy for the ordered MNIST task for adversarial perturbations. Again, the NRNNs show a superior resilience even to large perturbations, whereas the Antisymmetric and Exponential RNN appear to be sensitive even to small perturbations.

Figure 1 summarizes the performance of different models with respect to white noise and salt and pepper perturbations. The colored bands indicate $\pm 1$ standard deviation around the average performance. In all cases, the NRNN appears to be less sensitive to input perturbations as compared to the other models, while maintaining state-of-the-art performance for clean inputs.

## 7.2 Electrocardiogram (ECG) Classification

Next, we consider the Electrocardiogram (ECG) classification task that aims to discriminate between normal and abnormal heart beats of a patient that has severe congestive heart failure [20]. We use 500 sequences of length 140 for training, 500 sequences for validation, and 4000 sequences for testing.

Table 3 shows the average test accuracy (evaluated for models that are trained with 10 different seed values) for this task. We present results for additive white noise and multiplicative white noise perturbations. Here, the NRNN, trained with multiplicative noise level set to $0.03$ and additive noise levels set to $0.06$, performs best both on clean as well as on perturbed input sequences.

Figure 2 summarizes the performance of different models with respect to additive and multiplicative white noise perturbations. Again, the NRNN appears to be less sensitive to input perturbations as compared to the other models, while achieving state-of-the-art performance for clean inputs.

Table 2: Robustness w.r.t. adversarial perturbations on the ordered pixel-by-pixel MNIST task.

| Name | $r = 0.01$ | $r = 0.05$ | $r = 0.1$ | $r = 0.15$ |
|---|---|---|---|---|
| Antisymmetric RNN [7] | 79.4% | 24.7% | 11.4% | 10.2% |
| CoRNN [49] | 97.5% | 85.5% | 55.9% | 35.1% |
| Exponential RNN [32] | 94.5% | 59.3% | 19.7% | 14.3% |
| Lipschitz RNN [16] | 98.1% | 85.7% | 58.9% | 37.1% |
| NRNN (mult./add. noise: 0.02/0.02) | **98.8**% | 94.3% | 79.6% | 58.3% |
| NRNN (mult./add. noise: 0.02/0.05) | **98.8**% | **95.5**% | **86.8**% | **70.6**% |

Table 3: Robustness w.r.t. white ($\sigma$) and multiplicative ($\sigma_M$) noise perturbations on the ECG task.

| Name | clean | $\sigma = 0.4$ | $\sigma = 0.8$ | $\sigma = 1.2$ | $\sigma_M = 0.4$ | $\sigma_M = 0.8$ | $\sigma_M = 1.2$ |
|---|---|---|---|---|---|---|---|
| Antisymmetric RNN [7] | 97.1% | 96.6% | 91.6% | 77.0% | 96.6% | 94.6% | 91.2% |
| CoRNN [49] | 97.5% | 96.8% | 92.9% | 87.2% | 93.9% | 85.4% | 78.4% |
| Exponential RNN [32] | 97.4% | 95.6% | 86.4% | 76.7% | 95.7% | 89.4% | 81.3% |
| Lipschitz RNN [16] | **97.7**% | 97.4% | 95.1% | 88.9% | 97.6% | 97.0% | 95.6% |
| NRNN (mult./add. noise: 0.03/0.06) | **97.7**% | **97.5**% | **96.3**% | **92.6**% | **97.7**% | **97.3**% | **96.5**% |

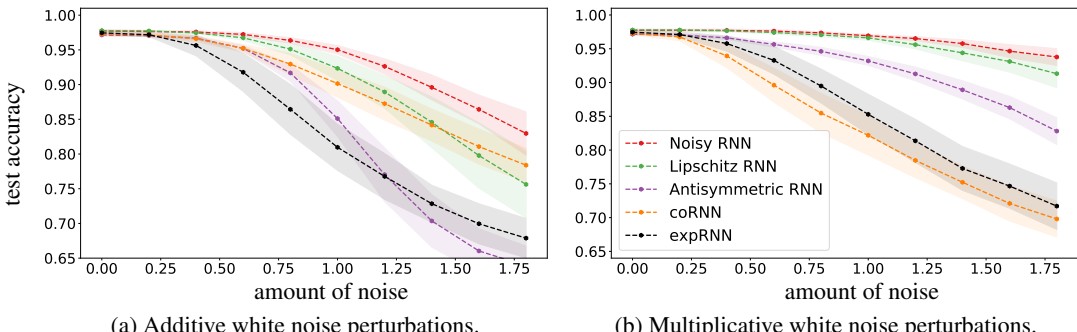

(a) Additive white noise perturbations.    (b) Multiplicative white noise perturbations.

Figure 2: Test accuracy for the ECG task as function of the strength of input perturbations.

## 8 Conclusion

In this paper we provide a thorough theoretical analysis of RNNs trained by injecting noise into the hidden states. Within the framework of SDEs, we study the regularizing effects of general noise injection schemes. The experimental results are in agreement with our theory and its implications, finding that Noisy RNNs achieve superior robustness to input perturbations, while maintaining state-of-the-art generalization performance. We believe our framework can be used to guide the principled design of a class of reliable and robust RNN classifiers.

Our work opens up a range of interesting future directions. In particular, for deterministic RNNs, it was shown that the models learn optimally near the edge of stability [8]. One could extend these analyses to NRNNs with the ultimate goal of improving their performance. On the other hand, as discussed in Section 5, although the noise is shown here to implicitly stabilize RNNs, it could negatively impact capacity for long-term memory [42, 70]. Providing analyses to account for this and the implicit bias due to the stochastic optimization procedure [50, 14] is the subject of future work.

## Acknowledgements

We are grateful for the generous support from Amazon AWS. S. H. Lim would like to acknowledge Nordita Fellowship 2018-2021 for providing support of this work. N. B. Erichson, L. Hodgkinson, and M. W. Mahoney would like to acknowledge the IARPA (contract W911NF20C0035), ARO, NSF, and and ONR via its BRC on RandNLA for providing partial support of this work. Our conclusions do not necessarily reflect the position or the policy of our sponsors, and no official endorsement should be inferred.

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
