# Supplementary Material for "Noisy Recurrent Neural Networks"

## A  Notation and Background

We begin by introducing some notations that will be used in this **SM**.

- $\|\cdot\|_F$ denotes Frobenius norm, $\|\cdot\|_p$ denote $p$-norm ($p > 0$) of a vector/matrix (in particular, $\|v\| := \|v\|_2$ denotes Euclidean norm of the vector $v$ and $\|A\|_2$ denotes the spectral norm of the matrix $A$).
- The $i$th element of a vector $v$ is denoted as $v^i$ or $[v]^i$ and the $(i,j)$-entry of a matrix $A$ is denoted as $A^{ij}$ or $[A]^{ij}$.
- $I$ denotes identity matrix (the dimension should be clear from the context).
- $\operatorname{tr}$ denotes trace, the superscript $T$ denotes transposition, and $\mathbb{R}^+ := (0, \infty)$.
- For a function $f : \mathbb{R}^n \to \mathbb{R}^m$ such that each of its first-order partial derivatives (with respect to $x$) exist on $\mathbb{R}^n$, $\frac{\partial f}{\partial x} \in \mathbb{R}^{m \times n}$ denotes the Jacobian matrix of $f$.
- For a scalar-valued function $g : \mathbb{R}^n \to \mathbb{R}$, $\nabla_h g$ denotes gradient of $g$ with respect to the variable $h \in \mathbb{R}^n$ and $H_h g$ denotes Hessian of $g$ with respect to $h$.
- The notation a.s. means $\mathbb{P}$-almost surely and $\mathbb{E}$ is expectation with respect to $\mathbb{P}$, where $\mathbb{P}$ is an underlying probability measure.
- For a matrix $M$, $M^{\mathrm{sym}} = (M + M^T)/2$ denote its symmetric part, $\lambda_{\min}(M)$ and $\lambda_{\max}(M)$ denote its minimum and maximum eigenvalue respectively, and $\sigma_{\min}(M)$ and $\sigma_{\max}(M)$ denote its minimum and maximum singular value respectively.
- For a vector $v = (v^1, \dots, v^d)$, $\operatorname{diag}(v)$ denotes the diagonal matrix with the $i$th diagonal entry equal $v^i$.
- $\mathbf{1}$ denotes a vector with all entries equal to one.
- $e_{ij}$ denotes the Kronecker delta.
- $\mathcal{C}(I; J)$ denotes the space of continuous $J$-valued functions defined on $I$.
- $\mathcal{C}^{2,1}(\mathcal{D} \times I; J)$ denotes the space of all $J$-valued functions $V(x, t)$ defined on $\mathcal{D} \times I$ which are continuously twice differentiable in $x \in \mathcal{D}$ and once differentiable in $t \in I$.

Next, we recall the RNN models considered in the main paper.

**Continuous-Time NRNNs.** For a terminal time $T > 0$ and an input signal $x = (x_t)_{t \in [0,T]} \in \mathcal{C}([0, T]; \mathbb{R}^{d_x})$, the output $y_t \in \mathbb{R}^{d_y}$, for $t \in [0, T]$, is a linear map of the hidden states $h_t \in \mathbb{R}^{d_h}$ satisfying the Itô stochastic differential equation (SDE):

$$\mathrm{d}h_t = f(h_t, x_t)\mathrm{d}t + \sigma(h_t, x_t)\mathrm{d}B_t, \qquad y_t = V h_t, \tag{23}$$

where $V \in \mathbb{R}^{d_y \times d_h}$, $f : \mathbb{R}^{d_h} \times \mathbb{R}^{d_x} \to \mathbb{R}^{d_h}$, $\sigma : \mathbb{R}^{d_h} \times \mathbb{R}^{d_x} \to \mathbb{R}^{d_h \times r}$ and $(B_t)_{t \geq 0}$ is an $r$-dimensional Wiener process.

In particular, as an example and for empirical experiments, we focus on the choice of drift function:

$$f(h, x) = Ah + a(Wh + Ux + b), \tag{24}$$

where $a : \mathbb{R} \to \mathbb{R}$ is a Lipschitz continuous scalar activation function (such as $\tanh$) extended to act on vectors pointwise, $A, W \in \mathbb{R}^{d_h \times d_h}$, $U \in \mathbb{R}^{d_h \times d_x}$ and $b \in \mathbb{R}^{d_h}$, and the choice of diffusion coefficient:

$$\sigma(h, x) = \epsilon(\sigma_1 I + \sigma_2 \text{diag}(f(h, x))), \tag{25}$$

where the noise level $\epsilon > 0$ is small, and $\sigma_1 \geq 0$ and $\sigma_2 \geq 0$ are tunable parameters describing the relative strength of additive noise and a multiplicative noise respectively.

We consider the following NRNN models by discretizing the SDE (23), as discussed in detail in the main paper.

**Discrete-Time NRNNs.** Let $0 := t_0 < t_1 < \cdots < t_M := T$ be a partition of the interval $[0, T]$. Denote $\delta_m := t_{m+1} - t_m$ for each $m = 0, 1, \ldots, M - 1$, and $\delta := (\delta_m)$. The Euler-Mayurama (E-M) scheme provides a family (parametrized by $\delta$) of approximations to the solution of the SDE in (23):

$$h_{m+1}^\delta = h_m^\delta + f(h_m^\delta, \hat{x}_m)\delta_m + \sigma(h_m^\delta, \hat{x}_m)\sqrt{\delta_m}\xi_m, \tag{26}$$

for $m = 0, 1, \ldots, M - 1$, where $(\hat{x}_m)_{m=0,\ldots,M-1}$ is a given sequential data, the $\xi_m \sim \mathcal{N}(0, I)$ are independent $r$-dimensional standard normal random vectors, and $h_0^\delta = h_0$. Eq. (26) describes the update equation of our NRNN models, an example of which is when $f$ and $\sigma$ are taken to be (24) and (25) respectively (see also the experiments in the main paper). In the special case when $\epsilon := 0$ in this example, we recover the Lipschitz RNN of [6].

It is worth mentioning that while higher-order integrators are also possible to consider, the presence of Itô white noise poses a significant challenge over the standard ODE case. Generally speaking, implementations of higher-order schemes require additional computational effort which may outweigh the benefit of using them. For instance, in implicit E-M schemes the zero of a nonlinear equation has to be determined in each time step [12]. In Milstein and stochastic Runge-Kutta schemes, there is an extra computational cost in simulating the Lévy area [21]. Similar challenges arise for other multistep schemes and higher-order schemes.

**Organizational Details.** This **SM** is organized as follows.

- In Section B, we provide results that guarantee existence and uniqueness of solutions to the SDE defining our continuous-time NRNNs.

- In Section C, we provide results that guarantee stability and convergence of our discrete-time NRNNs. These results are in fact very general and may be of independent interest.

- In Section D, we provide results on implicit regularization due to noise injection in both continuous-time and discrete-time NRNNs, in particular the proof of Theorem 1 in the main paper.

- In Section E, we provide some background and results to study classification margin and generalization bound of the corresponding discrete-time deterministic RNNs, in particular the proof of Theorem 2 in the main paper.

- In Section F, we discuss stability of continuous-time NRNNs and the noise-induced stabilization phenomenon, and provide conditions that guarantee almost sure exponential stability of the NRNNs, in particular the proof of Theorem 3 in the main paper.

- In Section G, we provide details on the empirical results in the main paper and additional results.

## B   Existence and Uniqueness of Solutions

Essential to any discussion concerning SDEs is the existence and uniqueness of solutions — in this case, we are interested in strong solutions [14].

In the following, we fix a complete filtered probability space $(\Omega, \mathcal{F}, (\mathcal{F}_t)_{t \geq 0}, \mathbb{P})$ which satisfies the usual conditions [14] and on which there is defined an $r$-dimensional Wiener process $(B_t)_{t \geq 0}$. We also fix a $T > 0$ and denote $f(h_t, t) := f(h_t, x_t)$, $\sigma(h_t, t) := \sigma(h_t, x_t)$ to emphasize the explicit dependence of the functions on time $t$ through the input $x_t$.

We start with the following assumptions on the SDE (23).

**Assumption B.** (a) (Global Lipschitz condition) The coefficients $f$ and $\sigma$ are $L$-Lipschitz, i.e., there exists a constant $L > 0$ such that

$$\|f(h,t) - f(h',t)\| + \|\sigma(h,t) - \sigma(h',t)\|_F \le L\|h - h'\| \tag{27}$$

for all $h, h' \in \mathbb{R}^{d_h}$ and $t \in [0,T]$.
(b) (Linear growth condition) $f$ and $\sigma$ satisfy the following linear growth condition, i.e., there exists a constant $K > 0$ such that

$$\|f(h,t)\|^2 + \|\sigma(h,t)\|_F^2 \le K(1 + \|h\|^2) \tag{28}$$

for all $h \in \mathbb{R}^{d_h}$ and $t \in [0,T]$.

Under Assumption B, it is a standard result from stochastic analysis that the SDE (23) has a unique solution (which is a continuous and adapted process $(h_t)_{t \in [0,T]}$ satisfying the integral equation $h_t = h_0 + \int_0^t f(h_s, s)ds + \int_0^t \sigma(h_s, s)dB_s$) for every initial value $h_0 \in \mathbb{R}^{d_h}$, for $t \in [0,T]$ (see, for instance, Theorem 3.1 in Section 2.3 of [23]). The uniqueness is in the sense that for any other solution $h'_t$ satisfying the SDE,

$$\mathbb{P}[h_t = h'_t \text{ for all } t \in [0,T]] = 1. \tag{29}$$

For our purpose, the following conditions suffice to satisfy Assumption B.

**Assumption C.** The function $a : \mathbb{R} \to \mathbb{R}$ is an activation function (i.e., a non-constant and Lipschitz continuous function), and $\sigma$ is $L_\sigma$-Lipschitz for some $L_\sigma > 0$.

**Lemma 1.** *Consider the SDE* (23) *defining our CT-NRNN. Then, under Assumption C, Assumption B is satisfied.*

*Proof.* Note that $f(h,t) = Ah + a(Wh + Ux_t + b)$, where $a$ is an activation function. For any $t \in [0,T]$,

$$\|f(h,t) - f(h',t)\| \le \|A(h - h')\| + \|a(Wh + Ux_t + b) - a(Wh' + Ux_t + b)\| \tag{30}$$

$$\le \|A\|\|h - h'\| + L_a\|W(h - h')\| \tag{31}$$

$$\le (\|A\| + L_a\|W\|)\|h - h'\|, \tag{32}$$

for all $h, h' \in \mathbb{R}^{d_h}$, where $L_a > 0$ is the Lipschitz constant of the (non-constant) activation function $a$. Therefore, the condition (a) in Assumption B is satisfied since by our assumption $\sigma$ is $L_\sigma$-Lipschitz for some constant $L_\sigma > 0$. In this case one can take $L = \max(\|A\| + L_a\|W\|, L_\sigma)$ in Eq. (27).

Since $f$ and $\sigma$ are $L$-Lipschitz, they satisfy the linear growth condition (b) in Assumption B. Indeed, if $f$ is $L$-Lipschitz, then for $t \in [0,T]$,

$$\|f(h,t)\| = \|f(h,t) - f(0,t) + f(0,t)\| \le L\|h\| + \|f(0,t)\| \le L\|h\| + C_f, \tag{33}$$

for some constant $C_f \in (0, \infty)$, where we have used the fact that $f(0,t) = a(Ux_t + b)$ is bounded for $t \in [0,T]$ (since continuous functions on compact sets are bounded). So,

$$\|f(h,t)\|^2 \le (L\|h\| + C_f)^2 \le L^2\|h\|^2 + 2LC_f\|h\| + C_f^2. \tag{34}$$

For $\|h\| \ge 1$, we have:

$$\|f(h,t)\|^2 \le (L^2 + 2LC_f)\|h\|^2 + C_f^2 \le (L^2 + 2LC_f + C_f^2)(1 + \|h\|^2). \tag{35}$$

For $\|h\| < 1$, we have:

$$\|f(h,t)\|^2 \le (L^2 + 2LC_f)\|h\| + C_f^2 \le (L^2 + 2LC_f) + C_f^2 \le (L^2 + 2LC_f + C_f^2)(1 + \|h\|^2). \tag{36}$$

Choosing $K = L^2 + 2LC_f + C_f^2$ gives us the linear growth condition for $f$.

Similarly, one can show that $\sigma$ satisfies the linear growth condition. The proof is done. $\square$

Throughout the paper, we work with SDEs satisfying Assumption C. The following additional assumption on the SDEs will be needed and invoked.

**Assumption D.** For $t \in [0, T]$, the partial derivatives of the coefficients $f^i(h, t)$, $\sigma^{ij}(h, t)$ with respect to $h$ up to order three (inclusive) exist. Moreover, the coefficients $f^i(h, t)$, $\sigma^{ij}(h, t)$ and all these partial derivatives are:

   (i) bounded and Borel measurable in $t$, for fixed $h \in \mathbb{R}^{d_h}$;

   (ii) Lipschitz continuous in $h$, for fixed $t \in [0, T]$.

In particular, Assumption D implies that these partial derivatives (with respect to $h$) of $f$ and $\sigma$ satisfy (a)-(b) in Assumption B. Assumption D holds for SDEs with commonly used activation functions such as hyperbolic tangent. We remark that Assumption C-D may be weakened in various directions (for instance, to locally Lipschitz coefficients) but for the purpose of this paper we need not go beyond these assumptions.

## C   Stability and Convergence of the Euler-Maruyama Schemes

We provide stability and strong convergence results for the explicit Euler-Mayurama (E-M) approximations of the SDE (23), which is time-inhomogeneous due to the dependence of the drift and possibly diffusion coefficient on a time-varying input, here. Intuitively, strong convergence results ensure that the approximated path follows the continuous path accurately, in contrast to weak convergence results which can only guarantee this at the level of probability distribution. The latest version of strong convergence results for time-homogeneous SDEs can be found in [7, 8]. The results for our time-inhomogeneous SDEs can be obtained by adapting the proof in [7] without much difficulty. Since we cannot find them in the literature, we provide them in this section.

First, we recall the discretization scheme. Let $0 := t_0 < t_1 < \cdots < t_M := T$ and $t_{m+1} = t_m + \delta_m$, for $m = 0, 1, \ldots, M - 1$ and some time step $\delta_m > 0$. Note that we work at full generality here since the step sizes $\delta_m$ are not necessarily uniform and may even depend on the numerical solution, i.e., $\delta_m = \delta(h_m^\delta)$ (see Example 1). The general results will be of independent interest, in particular for further explorations in designing other variants of NRNNs.

For $m = 0, 1, \ldots, M - 1$, consider

$$h_{m+1}^\delta = h_m^\delta + f(h_m^\delta, \hat{x}_m)\delta_m + \sigma(h_m^\delta, \hat{x}_m)\Delta B_m, \tag{37}$$

where $\Delta B_m := B_{t_{m+1}} - B_{t_m}$, $(\hat{x}_m)_{m=0,1,\ldots,M-1}$ is a given input sequential data, and $h_0^\delta = h_0$.

Let $\underline{t} = \max\{t_m : t_m \leq t\}$, $m_t = \max\{m : t_m \leq t\}$ for the nearest time point before time $t$, and its index. Denote the piecewise constant interpolant process $\bar{h}_t = h_{\underline{t}}^\delta$. It is convenient to use continuous-time approximations, so we consider the continuous interpolant that satisfies:

$$h_t^\delta = h_{\underline{t}}^\delta + f(h_{\underline{t}}^\delta, x_{\underline{t}})(t - \underline{t}) + \sigma(h_{\underline{t}}^\delta, x_{\underline{t}})(B_t - B_{\underline{t}}), \tag{38}$$

so that $h_t^\delta$ is the solution of the SDE:

$$dh_t^\delta = f(h_{\underline{t}}^\delta, x_{\underline{t}})dt + \sigma(h_{\underline{t}}^\delta, x_{\underline{t}})dB_t = f(\bar{h}_t, x_t)dt + \sigma(\bar{h}_t, x_t)dB_t. \tag{39}$$

We make the following assumptions about the time step.

**Assumption E.** The (possibly adaptive) time step function $\delta : \mathbb{R}^{d_h} \to \mathbb{R}^+$ is continuous and strictly positive, and there exist constants $\alpha, \beta > 0$ such that for all $h \in \mathbb{R}^{d_h}$, $\delta$ satisfies

$$\langle h, f(h, t) \rangle + \frac{1}{2}\delta(h)\|f(h, t)\|^2 \leq \alpha\|h\|^2 + \beta \tag{40}$$

for every $t \in [0, T]$.

Note that if another time step function $\delta^\epsilon(h)$ is smaller than $\delta(h)$, then $\delta^\epsilon(h)$ also satisfies Assumption E.

A simple adaptation of the proof of Theorem 2.1.1 in [7] to our case of time-inhomogeneous SDE gives the following result.

**Proposition 1** (Finite-time stability). *Under Assumption B and Assumption E, $T$ is a.s. attainable (i.e., for $\omega \in \Omega$, $\mathbb{P}[\exists N(\omega) < \infty \text{ s.t. } t_{N(\omega)} \geq T] = 1$) and for all $p > 0$ there exists a constant $C > 0$ (depending on only $p$ and $T$) such that*

$$\mathbb{E}\left[\sup_{t \in [0,T]} \|h_t^\delta\|^p\right] \leq C. \tag{41}$$

This is the discrete-time analogue of the result that $\mathbb{E}\left[\sup_{t \in [0,T]} \|h_t\|^p\right] < \infty$ for all $p > 0$, which can be proven by simply adapting the proof of Lemma 2.1.1. in [7].

In the case where the time step is adaptive, we take the following lower bound on the time step to bound the expected number of time steps (how quickly $\delta(h) \to 0$ as $\|h\| \to 0$).

**Assumption F.** There exist constants $a, b, q > 0$ such that the adaptive time step function satisfies:

$$\delta(h) \geq \frac{1}{a\|h\|^q + b}. \tag{42}$$

Next, we provide strong convergence result for the numerical approximation with the time step $\delta$. When the time step $\delta$ is adaptive, one needs to rescale the time step function by a small scalar-valued magnitude $\epsilon > 0$ and then consider the limit as $\epsilon \to 0$. Following [7], we make the following assumption.

**Assumption G.** The rescaled time step function $\delta^\epsilon$ satisfies

$$\epsilon \min(T, \delta(h)) \leq \delta^\epsilon(h) \leq \min(\epsilon T, \delta(h)), \tag{43}$$

where $\delta$ satisfies Assumption E-F.

Under this additional assumption, we have the following convergence result, which can be proven by adapting the proof of Theorem 2.1.2 in [7] to our time-inhomogeneous SDE case. The proof is based on the argument used for the uniform time step analysis (see Theorem 2.2 in [11]), taking into account the adaptive nature of the time step appropriately.

**Theorem 4** (Strong convergence). *Let the SDE (23) satisfy Assumption B and the time step function satisfy Assumption G. Then, for all $p > 0$,*

$$\lim_{\epsilon \to 0} \mathbb{E}\left[\sup_{t \in [0,T]} \|h_t^{\delta^\epsilon} - h_t\|^p\right] = 0, \tag{44}$$

*where $h_t^\delta$ is the continuous interpolant satisfying (38) and $h_t$ satisfies the SDE (23).*

In particular, the non-adaptive time stepping scheme satisfies the above assumptions. Therefore, stability and strong convergence of the schemes are guaranteed by the above results.

Under stronger assumptions on the drift $f$ we can obtain the order of strong convergence for the numerical schemes; see Theorem 2.1.3 in [7] for the case of time-homogeneous SDEs. This result can be adapted to our case to obtain order-$\frac{1}{2}$ strong convergence, which is also obtained in the special case when the step sizes are uniform (see Theorem 10.2.2 in [16]).

**Theorem 5** (Strong convergence rate). *Assume that $f$ satisfies the following one-sided Lipschitz condition, i.e., there exists a constant $\alpha > 0$ such that for all $h, h' \in \mathbb{R}^{d_h}$,*

$$\langle h - h', f(h,t) - f(h',t) \rangle \leq \alpha \|h - h'\|^2 \tag{45}$$

*for all $t \in [0,T]$, and the following locally polynomial growth Lipschitz condition, i.e., there exists $\gamma, \mu, q > 0$ such that for all $h, h' \in \mathbb{R}^{d_h}$,*

$$\|f(h,t) - f(h',t)\| \leq (\gamma(\|h\|^q + \|h'\|^q) + \mu)\|h - h'\|, \tag{46}$$

*for all $t \in [0,T]$. Moreover, assume that $\sigma$ is globally Lipschitz and the time step function satisfies Assumption G. Then, for all $p > 0$, there exists a constant $C > 0$ such that*

$$\mathbb{E}\left[\sup_{t \in [0,T]} \|h_t^{\delta^\epsilon} - h_t\|^p\right] \leq C\epsilon^{p/2}. \tag{47}$$

Lastly, it is worth mentioning the following adaptive scheme, which may be a useful option when designing NRNNs.

**Example 1** (Adaptive E-M). Under the same setup as the classical E-M setting, we may also introduce an adaptive step size scheme through a sequence of random vectors $d_m$. In this case,

$$h_{m+1}^\delta = h_m^\delta + d_m \odot f(h_m^\delta, \hat{x}_m)\Delta t_m + \sigma(h_m^\delta, \hat{x}_m)(\Delta t_m)^{1/2}\xi_m, \tag{48}$$

where $\odot$ denotes the pointwise (Hadamard) product, and each $d_n$ may be dependent on $h_i^\delta$ for $i \leq m$. Provided that $d_m \to (1, \dots, 1)$ uniformly almost surely as $\Delta t \to 0$, one could also obtain the same convergence as the classical E-M case. The adaptive setting allows for potentially better approximations by shrinking step sizes in places where the solution changes rapidly. An intuitive explanation for the instability of the standard E-M approximation of SDEs is that there is always a very small probability of a large Brownian increment which causes the approximation to produce a solution with undesirable growth. Using an adaptive time step eliminates this problem. Moreover, this scheme includes, in appropriate sense, the stochastic depth in [20] (see page 9 there) and the dropout in [19] as special cases upon choosing an appropriate $d_m$ and $\sigma$.

In particular, one can consider the following drift-tamed E-M scheme, where all components, $d_m^i$, of the elements of the sequence are generated as a function of $h_m^\delta$, i.e.,

$$d_m = \frac{1}{\max\{1, c_1\|h_m^\delta\| + c_2\}}\mathbf{1}, \tag{49}$$

for some $c_1, c_2 > 0$. In this way, the drift term is "tamed" by a solution-dependent multiplicative factor no larger than one, which prevents the hidden state in the next time step from becoming too large. This adaptive scheme is related to the one introduced in [13] to provide an explicit numerical method that would display strong convergence in circumstances where the standard E-M method does not. Under certain conditions strong convergence of this scheme can be proven (even for SDEs with superlinearly growing drift coefficients). Other adaptive schemes include the increment-tamed scheme of [12] and many others.

## D   Implicit Regularization in NRNNs

As discussed in the main paper, although the learning is carried out in discrete time, it is worth studying the continuous-time setting. The results for the continuous-time case may provide alternative perspectives and, more importantly, will be useful as a reference for exploring other discretization schemes for the CT-NRNNs. In Subsection D.1, we study implicit regularization for the continuous-time NRNNs. In Subsection D.2, we study implicit regularization for discrete-time NRNNs and comment on the difference between the continuous-time and discrete-time case.

We remark that the approach presented here is standard in showing implicit regularization. The essence of the approach is to view NRNN as a training scheme for the deterministic RNN. Also, note that had one attempted to conduct an analysis based on NRNN directly, the resulting bound would be stochastic due to the presence of the diffusion term and it is not clear how this bound helps explaining implicit regularization.

### D.1   Continuous-Time Setting

**Main Result and Discussions.** For the sake of brevity, we denote $f_t(\cdot) := f(\cdot, x_t)$ and $\sigma_t(\cdot) := \sigma(\cdot, x_t)$ for $t \in [0, T]$ in the following.

To begin, consider the process $(\bar{h}_t)_{t \in [0,T]}$ satisfying the following initial value problem (IVP):

$$d\bar{h}_t = f_t(\bar{h}_t)dt, \quad \bar{h}_0 = h_0. \tag{50}$$

Let $\Psi$ denote the unique fundamental matrix satisfying the following properties: for $0 \leq s \leq u \leq t \leq T$,

(a)

$$\frac{\partial \Psi(t,s)}{\partial t} = \frac{\partial f_t}{\partial \bar{h}}(\bar{h}_t)\Psi(t,s); \quad \frac{\partial \Psi(t,s)}{\partial s} = -\Psi(t,s)\frac{\partial f_t}{\partial \bar{h}}(\bar{h}_t); \tag{51}$$

(b) $\Psi(t, s) = \Psi(t, u)\Psi(u, s)$;

(c) $\Psi(t, s) = \Psi^{-1}(s, t)$;

(d) $\Psi(s, s) = I$.

Also, let $\Sigma(t, s) := \Psi(t, s)\sigma_s(\bar{h}_s)$ for $0 \le s \le t \le T$.

The following result links the expected loss function used for training CT-NRNNs to that for training deterministic CT-RNNs when the noise amplitude is small.

**Theorem 6** (Explicit regularization induced by noise injection for CT-NRNNs). *Under Assumption A in the main paper,*

$$\mathbb{E}\ell(h_T) = \ell(\bar{h}_T) + \frac{\epsilon^2}{2}[Q(\bar{h}) + R(\bar{h})] + \mathcal{O}(\epsilon^3), \tag{52}$$

*as $\epsilon \to 0$, where $Q$ and $R$ are given by*

$$Q(\bar{h}) = (\nabla l(\bar{h}_T))^T \int_0^T ds\, \Psi(T, s) \int_0^s du\, v(u) + (\nabla l(\bar{h}_T))^T \int_0^T ds\, w(s), \tag{53}$$

$$R(\bar{h}) = \int_0^T ds\, \mathrm{tr}(\Sigma(T, s)\Sigma(T, s)^\top \nabla^2 \ell(\bar{h}_T)), \tag{54}$$

*with $v(u)$ a vector with the pth component ($p = 1, 2, \ldots, d_h$):*

$$v^p(u) = tr(\Sigma(s, u)\Sigma^T(s, u)\nabla^2[f_s]^p(\bar{h}_s)), \tag{55}$$

*and $w(s)$ a vector with the qth component ($q = 1, 2, \ldots, d_h$):*

$$w^q(s) = \sum_{k=1}^r \sum_{j,l=1}^{d_h} \Psi_{k,l}^{qj}(T, s)\partial_l \sigma_s^{jk}(\bar{h}_s)\sigma_s^{lk}(\bar{h}_s). \tag{56}$$

Therefore, to study the difference between the CT-NRNNs and their deterministic version, it remains to investigate the role of $Q$ and $R$ in Theorem 6. If the Hessian is positive semi-definite, then $R(\bar{h})$ is also positive semi-definite and thus a viable regularizer. On the other hand, $Q(\bar{h})$ need not be non-negative. However, by assuming that $\nabla^2 f$ and $\nabla \sigma^{ij}$ are small (that is, $f$ is approximately linear and $\sigma$ relatively independent of $\bar{h}$), then $Q$ can be perceived negligible and we may focus predominantly on $R$. An argument of this kind was used in [4] in the context of Gauss-Newton Hessian approximations. In particular, $Q = 0$ for linear NRNNs with additive noise. Therefore, Theorem 6 essentially tells us that injecting noise to deterministic RNN is approximately equivalent to considering a regularized objective functional. Moreover, the explicit regularizer is solely determined by the flow generated by the Jacobian $\frac{\partial f_t}{\partial \bar{h}}(\bar{h}_t)$, the diffusion coefficient $\sigma_t$ and the Hessian of the loss function, all evaluated along the dynamics of the deterministic RNN.

Under these assumptions, ignoring higher-order terms and bounding the Frobenius inner product in (54), we can interpret training with CT-NRNN as an approximation of the following optimal control problem [29] with the running cost $C(t) := \frac{1}{2}\mathrm{tr}(\sigma_t(\bar{h}_t)^T \Psi(T, t)^T \nabla^2 \ell(\bar{h}_t)\Psi(T, t)\sigma_t(\bar{h}_t))$:

$$\min \mathbb{E}_{(\boldsymbol{x}, y) \sim \mu} \left[ \ell(\bar{h}_T) + \epsilon^2 \int_0^T C(t)dt \right] \tag{57}$$

$$\text{s.t. } d\bar{h}_t = f_t(\bar{h}_t)dt, \ t \in [0, T], \ \bar{h}_0 = h_0, \tag{58}$$

where $(\boldsymbol{x} := (x_t)_{t \in [0, T]}, y)$ denotes a training example drawn from the distribution $\mu$ and the minimization is with respect to the parameters (controls) in the corresponding deterministic RNN. On the other hand, we can interpret training with the deterministic RNN as the above optimal control problem with zero running cost or regularization. Note that if the Hessian matrix is symmetric positive semi-definite, then $C(t)$ is a quadratic form with the associated metric tensor $M_t^T M_t := \nabla^2 \ell(\bar{h}_t)$ and

$$C(t) = \frac{1}{2}\langle \Psi(T, t)\sigma_t, \Psi(T, t)\sigma_t \rangle_{M_t} = \frac{1}{2}\|M_t \Psi(T, t)\sigma_t\|_F^2 \le \frac{1}{2}\|\sigma_t\|_F^2 \|M_t\|_F^2 \|\Psi(T, t)\|_F^2. \tag{59}$$

Overall, we can see that the use of NRNNs as a regularization mechanism reduces the fundamental matrices $\Psi(T, s)$ according to the magnitude of the elements of $\sigma_t$.

**Proof of Theorem 6.** Next, we prove Theorem 6. We will need some auxiliary results before doing so.

For a small perturbation parameter $\epsilon > 0$, the hidden states now satisfy the SDE

$$\mathrm{d}h_t = f_t(h_t)\mathrm{d}t + \epsilon\sigma_t(h_t)\mathrm{d}B_t,$$

where we have used the shorthand $f_t(\cdot) = f(\cdot, x_t)$ and $\sigma_t(\cdot) = \sigma(\cdot, x_t)$. To investigate the effect of the perturbation, consider the following hierarchy of differential equations:

$$\mathrm{d}h_t^{(0)} = f_t(h_t^{(0)})\mathrm{d}t, \tag{60}$$

$$\mathrm{d}h_t^{(1)} = \frac{\partial f_t}{\partial h}(h_t^{(0)})h_t^{(1)}\mathrm{d}t + \sigma_t(h_t^{(0)})\mathrm{d}B_t, \tag{61}$$

$$\mathrm{d}h_t^{(2)} = \frac{\partial f_t}{\partial h}(h_t^{(0)})h_t^{(2)}\mathrm{d}t + \Phi_t^{(1)}(h_t^{(0)}, h_t^{(1)})\mathrm{d}t + \Phi_t^{(2)}(h_t^{(0)}, h_t^{(1)})\mathrm{d}B_t, \tag{62}$$

with $h_0^{(0)} = h_0$, $h_0^{(1)} = 0$, and $h_0^{(2)} = 0$, and where

$$\Phi_t^{(1)}(h_0, h_1) = \frac{1}{2}\sum_{i,j}\frac{\partial^2 f_t}{\partial h^i \partial h^j}(h_0)h_1^i h_1^j \tag{63}$$

$$\Phi_t^{(2)}(h_0, h_1) = \sum_i \frac{\partial \sigma_t}{\partial h^i}(h_0)h_1^i. \tag{64}$$

In the sequel, we will suppose Assumption A in the main paper (which is equivalent to Assumption B and Assumption D) holds. Under this assumption, each of these initial value problems have a unique solution for $t \in [0, T]$. The processes $h_t^{(0)}$, $h_t^{(1)}$ and $h_t^{(2)}$ denote the zeroth-, first-, and second-order terms in an expansion of $h_t$ about $\epsilon = 0$. This can be easily seen using Kunita's theory of stochastic flows. In particular, by Theorem 3.1 in [17], letting $h_{\epsilon,t}^{(1)} = \frac{\partial h_t}{\partial \epsilon}$, we find that

$$\mathrm{d}h_{\epsilon,t}^{(1)} = \frac{\partial f_t}{\partial h}(h_t)h_{\epsilon,t}^{(1)}\mathrm{d}t + \left(\sigma_t(h_t) + \epsilon\Phi_t^{(2)}(h_t, h_{\epsilon,t}^{(1)})\right)\mathrm{d}B_t,$$

and so we find that $h_{0,t}^{(1)} = h_t^{(1)}$. Similarly, $h_{\epsilon,t}^{(2)} = \frac{\partial^2 h_t}{\partial \epsilon^2} = \frac{\partial h_{\epsilon,t}^{(1)}}{\partial \epsilon}$ can be shown to satisfy

$$\mathrm{d}h_{\epsilon,t}^{(2)} = \frac{\partial f_t}{\partial h}(h_t)h_{\epsilon,t}^{(2)}\mathrm{d}t + 2\Phi_t^{(1)}(h_t, h_{\epsilon,t}^{(1)})\mathrm{d}t$$

$$+ \left(2\Phi_t^{(2)}(h_t, h_{\epsilon,t}^{(1)}) + \epsilon\sum_k [h_{\epsilon,t}^{(1)}]^k \frac{\partial}{\partial h^k}\Phi_t^{(2)}(h_t, h_{\epsilon,t}^{(1)})\right)\mathrm{d}B_t.$$

This equation is obtained by applying Theorem 3.1 in [17] to find the first derivative of the system $(h_t, h_{\epsilon,t}^{(1)})$ with respect to $\epsilon$ and projecting to the second coordinate. Taking $\epsilon = 0$, we find that $h_{0,t}^{(2)} = 2h_t^{(2)}$. Therefore, informally, a pathwise second-order Taylor expansion about $\epsilon = 0$ reveals that $h_t = h_t^{(0)} + \epsilon h_t^{(1)} + \epsilon^2 h_t^{(2)} + \mathcal{O}(\epsilon^3)$. To formalize this statement, we will later bound the third-order error term in Lemma 3.

While the equation for $h_t^{(0)}$ is not explicitly solvable, both $h_t^{(1)}$ and $h_t^{(2)}$ are. In particular, for $t \in [0, T]$ (see Eq. (4.28) in [25]):

$$h_t^{(1)} = \int_0^t \Psi(t, s)\sigma_s(h_s^{(0)})\mathrm{d}B_s = \int_0^t \Sigma(t, s)\mathrm{d}B_s, \tag{65}$$

$$h_t^{(2)} = \int_0^t \Psi(t, s)\Phi_s^{(1)}(h_s^{(0)}, h_s^{(1)})\mathrm{d}s + \int_0^t \Psi(t, s)\Phi_s^{(2)}(h_s^{(0)}, h_s^{(1)})\mathrm{d}B_s. \tag{66}$$

The key result needed to prove Theorem 6 is contained in the following theorem. In the sequel, big $\mathcal{O}$ notation is to be understood in the almost sure sense.

**Theorem 7.** *For a scalar-valued loss function $\ell \in \mathcal{C}^2(\mathbb{R}^{d_h})$, for $t \in [0, T]$,*

$$\ell(h_t) = \ell(h_t^{(0)}) + \epsilon\nabla\ell(h_t^{(0)}) \cdot h_t^{(1)} + \epsilon^2\left(\nabla\ell(h_t^{(0)}) \cdot h_t^{(2)} + \frac{1}{2}(h_t^{(1)})^\top \nabla^2\ell(h_t^{(0)})(h_t^{(1)})\right) + \mathcal{O}(\epsilon^3),$$

*as $\epsilon \to 0$.*

We now prove Theorem 7. The proof relies on two lemmas. The first bounds the solutions $h^{(i)}$ over $[0, T]$.

**Lemma 2.** *For any $p > 0$, $\sup_{s \in [0,T]} \|h_s^{(0)}\|^p < \infty$ and $\mathbb{E} \sup_{s \in [0,T]} \|h_s^{(i)}\|^p < \infty$ for $i = 1, 2$.*

*Proof.* For $s \in [0, T]$, $h_s^{(0)} = h_0 + \int_0^s f_u(h_u^{(0)}) \mathrm{d}u$, so recalling $(x+y)^2 \leq 2x^2 + 2y^2$ and $\|f_t(h)\|^2 \leq K(1 + \|h\|^2)$,

$$\|h_s^{(0)}\|^2 \leq 2\|h_0\|^2 + 2 \int_0^s \|f_u(h_u^{(0)})\|^2 \mathrm{d}u$$

$$\leq 2 \left( \|h_0\|^2 + K^2 s + K^2 \int_0^s \|h_u^{(0)}\|^2 \mathrm{d}u \right).$$

Therefore, by Gronwall's inequality,

$$\|h_s^{(0)}\|^2 \leq 2(\|h_0\|^2 + K^2 s) e^{2K^2 s}$$

$$\leq 2(\|h_0\|^2 + K^2 T) e^{2K^2 T} < +\infty,$$

and so $\sup_{s \in [0,T]} \|h_s^{(0)}\| < \infty$. Similarly, for $s \in [0, T]$,

$$h_s^{(1)} = \int_0^s \frac{\partial f_u}{\partial h}(h_u^{(0)}) h_u^{(1)} \mathrm{d}u + \int_0^s \sigma_u(h_u^{(0)}) \mathrm{d}B_u.$$

Therefore, for $p \geq 2$ (since $(x+y)^p \leq 2^{p-1}(x^p + y^p)$ by Jensen's inequality):

$$\|h_s^{(1)}\|^p \leq 2^{p-1} \int_0^s \left\| \frac{\partial f_u}{\partial h}(h_u^{(0)}) \right\|^p \|h_u^{(1)}\|^p \mathrm{d}u + 2^{p-1} \left\| \int_0^s \sigma_u(h_u^{(0)}) \mathrm{d}B_u \right\|^p.$$

Because the Itô integral is a continuous martingale, the Burkholder-Davis-Gundy inequality (see Theorem 3.28 in [14]) implies that for positive constants $C_p$ depending only on $p$ (but not necessarily the same in each appearance),

$$\mathbb{E} \sup_{s \in [0,T]} \|h_s^{(1)}\|^p \leq C_p \int_0^s \mathbb{E} \sup_{s \in [0,u]} \|h_s^{(1)}\|^p \mathrm{d}u + C_p \left( \int_0^t \|\sigma_u(h_u^{(0)})\|^2 \mathrm{d}u \right)^{p/2}$$

An application of Gronwall's inequality yields

$$\mathbb{E} \sup_{s \in [0,T]} \|h_s^{(1)}\|^p \leq C_p \left( \int_0^T \|\sigma_u(h_u^{(0)})\|^2 \mathrm{d}u \right)^{p/2} e^{C_p T}.$$

Therefore, $\mathbb{E} \sup_{s \in [0,T]} \|h_s^{(1)}\|^p < \infty$ for all $p \geq 2$. The $p \in (0, 2)$ case follows from Hölder's inequality. Repeating this same approach for $h_s^{(2)}$ completes the proof. $\square$

The second of our two critical lemmas provides a pathwise expansion of $h_t$ about $\epsilon$ in the vein of [3]. Doing so characterizes the response of the NRNN hidden states to small noise perturbations at the sample path level. It can be seen as a strengthening of Theorem 2.2 in [9] for our time-inhomogeneous SDEs.

**Lemma 3.** *For a fixed $\epsilon_0 > 0$, and any $0 < \epsilon \leq \epsilon_0$, with probability one,*

$$h_t = h_t^{(0)} + \epsilon h_t^{(1)} + \epsilon^2 h_t^{(2)} + \epsilon^3 R_3^\epsilon(t),$$

*where for any $p > 0$,*

$$\sup_{\epsilon \in (0,\epsilon_0)} \mathbb{E} \sup_{t \in [0,T]} \|R_3^\epsilon(t)\|^p < \infty. \tag{67}$$

*Proof.* It suffices to show that $\sup_{\epsilon \in (0,\epsilon_0)} \mathbb{E} \sup_{t \in [0,T]} \|R_3^\epsilon(t)\|^p < \infty$ for $p \geq 2$ — the $p \in (0, 2)$ case follows from Hölder's inequality. In the sequel, we shall let $K$ denote a finite number (not necessarily the same in each appearance) depending only on $f, \sigma, T, \epsilon_0$, and $p$, and therefore independent of $t, \epsilon$.

For $\epsilon > 0$, let $h_t^\epsilon = h_t^{(0)} + \epsilon h_t^{(1)} + \epsilon^2 h_t^{(2)}$ and $R_3(t) = \epsilon^{-3}(h_t - h_t^\epsilon)$, where $h_t, h_t^{(1)}, h_t^{(2)}$ are coupled together through the same Brownian motion. Then

$$\epsilon^3 R_3(t) = \int_0^t \left( f_s(h_s) - f_s(h_s^{(0)}) - \epsilon \frac{\partial f_s}{\partial h}(h_s^{(0)})h_s^{(1)} - \epsilon^2 \frac{\partial f_s}{\partial h}(h_s^{(0)})h_s^{(2)} - \epsilon^2 \Phi_s^{(1)}(h_s^{(0)}, h_s^{(1)}) \right) ds$$

$$+ \epsilon \int_0^t \sigma_s(h_s) - \sigma_s(h_s^{(0)}) - \epsilon \Phi_s^{(2)}(h_s^{(0)}, h_s^{(1)}) dB_s.$$

To simplify, we decompose $\epsilon^3 R_3(t)$ into the sum of four random variables $\theta_i(t)$, $i = 1, \ldots, 4$, given by

$$\theta_1(t) = \int_0^t [f_s(h_s) - f_s(h_s^\epsilon)] \, ds$$

$$\theta_2(t) = \int_0^t \left[ f_s(h_s^\epsilon) - f_s(h_s^{(0)}) - \epsilon \frac{\partial f_s}{\partial h}(h_s^{(0)})h_s^{(1)} - \epsilon^2 \frac{\partial f_s}{\partial h}(h_s^{(0)})h_s^{(2)} - \epsilon^2 \Phi_s^{(1)}(h_s^{(0)}, h_s^{(1)}) \right] ds$$

$$\theta_3(t) = \epsilon \int_0^t \left[ \sigma_s(h_s) - \sigma_s(h_s^{(0)} + \epsilon h_s^{(1)}) \right] dB_s$$

$$\theta_4(t) = \epsilon \int_0^t \left[ \sigma_s(h_s^{(0)} + \epsilon h_s^{(1)}) - \sigma_s(h_s^{(0)}) - \epsilon \Phi_s^{(2)}(h_s^{(0)}, h_s^{(1)}) \right] dB_s.$$

Beginning with the more straightforward terms $\theta_1(t), \theta_3(t)$, by Lipschitz continuity of $f$,

$$\|f_s(h_s) - f_s(h_s^\epsilon)\| \leq L_f \epsilon^3 \|R_3(s)\|,$$

and so

$$\mathbb{E} \sup_{s \in [0,t]} \|\theta_1(s)\|^p \leq K\epsilon^{3p} \int_0^t \mathbb{E} \sup_{s \in [0,u]} \|R_3(s)\| \, du.$$

In the same way, $\|\sigma_s(h_s) - \sigma_s(h_s^{(0)} + \epsilon h_s^{(1)})\| \leq L_\sigma \epsilon^2 \|h_s^{(2)} + \epsilon R_3(s)\|$. Recall that $(\int_0^t g(s) ds)^p \leq t^{p-1} \int_0^t g(s)^p ds$ by Jensen's inequality. Now, $\theta_3(s)$ is a continuous martingale, and hence, the Burkholder-Davis-Gundy inequality (see Theorem 3.28 in [14]) implies that for some constant $C_p > 0$ depending only on $p$,

$$\mathbb{E} \sup_{s \in [0,t]} \|\theta_3(s)\|^p \leq \epsilon^p C_p \left( \int_0^t \mathbb{E} \left\| \sigma_s(h_s) - \sigma_s(h_s^{(0)} + \epsilon h_s^{(1)}) \right\|^2 ds \right)^{p/2}$$

$$\leq C_p L_\sigma^p \epsilon^{3p} \left( \int_0^t \mathbb{E}\|h_s^{(2)} + \epsilon R_3(s)\|^2 ds \right)^{p/2}$$

$$\leq C_p L_\sigma^p \epsilon^{3p} T^{p/2-1} \int_0^t \mathbb{E}\|h_s^{(2)} + \epsilon R_3(s)\|^p ds$$

$$\leq C_p L_\sigma^p \epsilon^{3p} 2^{p-1} T^{p/2-1} \left( \int_0^t \mathbb{E}\|h_s^{(2)}\|^p ds + \epsilon^p \int_0^t \mathbb{E}\|R_3(s)\|^p ds \right).$$

From Lemma 2, it follows that

$$\mathbb{E} \sup_{s \in [0,t]} \|\theta_3(s)\|^p \leq K\epsilon^{3p} \left( 1 + \epsilon^p \int_0^t \mathbb{E} \sup_{s \in [0,u]} \|R_3(s)\|^p du \right).$$

Treating the $\theta_2$ term next, for each $s \in [0,t]$, by Taylor's theorem, there exists some $\epsilon_s \in (0, \epsilon)$ such that

$$f_s(h_s^\epsilon) - f_s(h_0) - \epsilon \frac{\partial f_s}{\partial h}(h_0)h_1 = \epsilon^2 \frac{\partial f_s}{\partial h}(h_s^{\epsilon_s})h_2 + \epsilon^2 \Phi_s^{(1)}(h_s^{\epsilon_s}, h_1).$$

Therefore, by Lipschitz continuity of the derivatives of $f$,

$$\theta_2(t) = \epsilon^2 \int_0^t \left( \frac{\partial f_s}{\partial h}(h_s^{\epsilon_s})h_2 + \Phi_s^{(1)}(h_s^{\epsilon_s}, h_1) - \frac{\partial f_s}{\partial h}(h_0)h_2 - \Phi_s^{(1)}(h_0, h_1) \right) ds$$

$$\leq K\epsilon^2 \int_0^t \|h_s^{\epsilon_s} - h_0\| ds$$

$$\leq K\epsilon^3 \int_0^t \|h_s^{(1)}\| + \epsilon \|h_s^{(2)}\| ds.$$

From Lemma 2, it follows that

$$\mathbb{E} \sup_{s \in [0,T]} \|\theta_2(s)\|^p \leq K\epsilon^{3p}.$$

Similarly, by Taylor's theorem, there exists $\epsilon_s \in (0, \epsilon)$ such that

$$\sigma_s(h_s^{(0)} + \epsilon h_s^{(1)}) - \sigma_s(h_s^{(0)}) = \epsilon \Phi_s^{(2)}(h_s^{(0)} + \epsilon_s h_s^{(1)}, h_s^{(1)}),$$

and so for $p \geq 2$, by the Burkholder-Davis-Gundy inequality and Lipschitz continuity of the derivatives of $\sigma$,

$$\mathbb{E} \sup_{s \in [0,t]} \|\theta_4(s)\|^p \leq C_p \epsilon^{2p} \left( \int_0^t \mathbb{E} \left\| \Phi_s^{(2)}(h_s^{(0)} + \epsilon_s h_s^{(1)}, h_s^{(1)}) - \Phi_s^{(2)}(h_s^{(0)}, h_s^{(1)}) \right\|^2 \mathrm{d}s \right)^{p/2}$$

$$\leq KC_p \epsilon^{3p} \left( \int_0^t \mathbb{E}\|h_s^{(1)}\|^2 \mathrm{d}s \right)^{p/2}$$

$$\leq KC_p T^{p/2} \epsilon^{3p} \mathbb{E} \sup_{s \in [0,T]} \|h_s^{(1)}\|^p \leq K\epsilon^{3p}.$$

Combining estimates for $\theta_1, \theta_2, \theta_3, \theta_4$,

$$\mathbb{E} \sup_{s \in [0,t]} \|R_3(s)\|^p$$

$$= 4^{p-1} \epsilon^{-3p} \left( \mathbb{E} \sup_{s \in [0,t]} \|\theta_1(s)\|^p + \mathbb{E} \sup_{s \in [0,t]} \|\theta_2(s)\|^p + \mathbb{E} \sup_{s \in [0,t]} \|\theta_3(s)\|^p + \mathbb{E} \sup_{s \in [0,T]} \|\theta_4(s)\|^p \right)$$

$$\leq K \left( 1 + \int_0^t \mathbb{E} \sup_{s \in [0,u]} \|R_3(s)\|^p \, \mathrm{d}u \right),$$

and so by Gronwall's inequality, $\mathbb{E} \sup_{s \in [0,t]} \|R_3(s)\|^p \leq Ke^{Kt}$. Since $K$ is independent of $t \leq T$ and $\epsilon \leq \epsilon_0$, it follows that

$$\sup_{\epsilon \in (0, \epsilon_0)} \mathbb{E} \sup_{s \in [0,t]} \|R_3(s)\|^p \leq Ke^{Kt} < +\infty,$$

and the result follows. $\qquad \square$

We remark that perturbative techniques such as the one used to obtain Theorem 3 are standard in the theory of stochastic flows.

Theorem 7 now follows in a straightforward fashion from Lemma 3 by taking a second-order Taylor expansion of $\ell(h_t^{(0)} + \epsilon h_t^{(1)} + \epsilon^2 h_t^{(2)} + \mathcal{O}(\epsilon^3))$ about $\epsilon = 0$.

We are now in a position to prove Theorem 6 using Theorem 7.

*Proof of Theorem 6.* From Theorem 7, we have, upon taking expectation:

$$\mathbb{E}\ell(h_t) = \ell(h_t^{(0)}) + \epsilon(\nabla_{h^{(0)}}\ell)^T \mathbb{E}h_t^{(1)} + \epsilon^2 \left( (\nabla_{h^{(0)}}\ell)^T \mathbb{E}h_t^{(2)} + \frac{1}{2}\mathbb{E}(h_t^{(1)})^T (H_{h^{(0)}}\ell) h_t^{(1)} \right) + \mathcal{O}(\epsilon^3),$$

(68)

for $t \in [0, T]$, as $\epsilon \to 0$, where $H_{h^{(0)}}$ denotes Hessian operator and the $h_t^{(i)}$ satisfy Eq. (60)-(62).

Since $\nabla f_t$ and its derivative are bounded and are thus Lipschitz continuous, by Picard's theorem the IVP has a unique solution. Moreover, it follows from our assumptions that the solution to the IVP is square-integrable (i.e., $\int_0^t \|\Psi(t, s)\|_F^2 ds < \infty$ for any $t \in [0, T]$). Therefore, the solution $h_t^{(1)}$ to Eq. (61) can be uniquely represented as the following Itô integral:

$$h_t^{(1)} = \int_0^t \Psi(t, s) \sigma(h_s^{(0)}, s) dB_s,$$

(69)

where $\Psi(t, s)$ is the (deterministic) fundamental matrix solving the IVP (51). We have $\mathbb{E}h_t^{(1)} = 0$ and

$$\mathbb{E}\|h_t^{(1)}\|^2 = \int_0^t \|\Psi(t, s) \sigma(h_s^{(0)}, s)\|_F^2 ds < \infty.$$

(70)

Similar argument together with Assumption D shows that the solution $h_t^{(2)}$ to Eq. (62) admits the following unique integral representation, with the $i$th component:

$$h_t^{(2)i} = \frac{1}{2}\int_0^t \Psi^{ij}(t,s)[h_s^{(1)}]^l \frac{\partial^2 b^j}{\partial[h_s^{(0)}]^l \partial[h_s^{(0)}]^k}[h_s^{(1)}]^k ds + \int_0^t \Psi^{ij}(t,s)\frac{\partial \sigma^{jk}}{\partial[h_s^{(0)}]^l}[h_s^{(1)}]^l dB_s^k, \quad (71)$$

where the last integral above is a uniquely defined Itô integral.

Plugging Eq. (69) into the above expression and then taking expectation, we have:

$$\mathbb{E}h_t^{(2)i} = \frac{1}{2}\mathbb{E}\int_0^t ds\,\Psi^{ij}(t,s)\frac{\partial^2 b^j}{\partial[h_s^{(0)}]^l \partial[h_s^{(0)}]^k}\int_0^s dB_{u_1}^{l_2}\int_0^s dB_{u_2}^{k_2}\Psi^{ll_1}(s,u_1)\sigma^{l_1 l_2}\sigma^{k_1 k_2}\Psi^{kk_1}(s,u_2)$$

$$+\frac{1}{2}\mathbb{E}\int_0^t dB_s^k \Psi^{ij}(t,s)\frac{\partial \sigma^{jk}}{\partial[h_s^{(0)}]^l}\int_0^t dB_u^{l_2}\Psi^{ll_1}(s,u)\sigma^{l_1 l_2}(h_u^{(0)},u), \quad (72)$$

where we have performed change of variable to arrive at the last double integral above.

Using the semigroup property of $\Psi$, we have $\Psi(t,s) = \Psi(t,0)\Psi^{-1}(s,0)$ for any $s \leq t$ (and so $\Psi^{ij}(t,s) = \Psi^{ij_1}(t,0)(\Psi^{-1})^{j_1 j}(s,0)$ and $\Psi^{ll_1}(s,u) = \Psi^{ll_2}(s,0)(\Psi^{-1})^{l_2 l_1}(u,0)$ etc.). Using this property in (72) and then evaluating the resulting expression using properties of moments of stochastic integrals (applying Eq. (5.7) and Proposition 4.16 in [10] – note that Itô isometry follows from Eq. (5.7) there), we obtain $(\nabla_{h^{(0)}}\ell)^T \mathbb{E}h_t^{(2)} = Q(h^{(0)})$, where $Q$ satisfies Eq. (53).

Similarly, plugging Eq. (69) into $\mathbb{E}h_t^{(1)T}(H_{h^{(0)}}\ell)h_t^{(1)}$, and then proceeding as above and applying the cyclic property of trace, give $\frac{1}{2}\mathbb{E}(h_t^{(1)})^T(H_{h^{(0)}}\ell)h_t^{(1)} = R(h^{(0)})$, where $R$ satisfies Eq. (54). The proof is done. $\qquad\square$

## D.2 Discrete-Time Setting: Proof of Theorem 1 in the Main Paper

The goal in this subsection is to prove Theorem 1 in the main paper, the discrete-time analogue of Theorem 6. We recall the theorem in the following.

**Theorem 8** (Explicit regularization induced by noise injection for discrete-time NRNNs – Theorem 1 in the main paper)**.** *Under Assumption A in the main paper,*

$$\mathbb{E}\ell(h_M^\delta) = \ell(\bar{h}_M^\delta) + \frac{\epsilon^2}{2}[\hat{Q}(\bar{h}^\delta) + \hat{R}(\bar{h}^\delta)] + \mathcal{O}(\epsilon^3), \quad (73)$$

*as $\epsilon \to 0$, where the terms $\hat{Q}$ and $\hat{R}$ are given by*

$$\hat{Q}(\bar{h}^\delta) = (\nabla l(\bar{h}_M^\delta))^T \sum_{k=1}^M \delta_{k-1}\hat{\Phi}_{M-1,k}\sum_{m=1}^{M-1}\delta_{m-1}\boldsymbol{v}_m, \quad (74)$$

$$\hat{R}(\bar{h}^\delta) = \sum_{m=1}^M \delta_{m-1}\mathrm{tr}(\sigma_{m-1}^T \hat{\Phi}_{M-1,m}^T H_{\bar{h}^\delta} l\,\hat{\Phi}_{M-1,m}\sigma_{m-1}), \quad (75)$$

*with $\boldsymbol{v}_m$ a vector with the $p$th component $(p = 1,\ldots,d_h)$:*

$$[v_m]^p = \mathrm{tr}(\sigma_{m-1}^T \hat{\Phi}_{M-2,m}^T H_{\bar{h}^\delta}[f_M]^p \hat{\Phi}_{M-2,m}\sigma_{m-1}).$$

*Moreover,*

$$|\hat{Q}(\bar{h}^\delta)| \leq C_Q \Delta^2, \quad |\hat{R}(\bar{h}^\delta)| \leq C_R\Delta, \quad (76)$$

*for $C_Q, C_R > 0$ independent of $\Delta$.*

To prove Theorem 8, the key idea is to first obtain a discretized version of the loss function in Theorem 7 by either discretizing the results in Theorem 7 or by proving directly from the discretized equations (26). It then remains to compute the expectation of this loss as functional of the discrete-time process. The first part is straightforward while the second part involves some tedious recursive computations.

Let $0 := t_0 < t_1 < \cdots < t_M := T$ be a partition of the interval $[0,T]$ and let $\delta_m = t_{m+1} - t_m$ for each $m = 0, 1,\ldots,M-1$. For small parameter $\epsilon > 0$, the E-M scheme is given by:

$$h_{m+1}^\delta = h_m^\delta + f(h_m^\delta,\hat{x}_m)\delta_m + \epsilon\sigma(h_m^\delta,\hat{x}_m)\sqrt{\delta_m}\xi_m, \quad (77)$$

where $(\hat{x}_m)_{m=0,\ldots,M-1}$ is a given sequential data, each $\xi_m \sim \mathcal{N}(0, I)$ is an independent $r$-dimensional standard normal random vector, and $h_0^\delta = h_0$.

Consider the following hierarchy of recursive equations. For the sake of notation cleanliness, we replace the superscript $\delta$ by hat when denoting the $\delta$-dependent approximating solutions in the following.

For $m = 0, 1, \ldots, M - 1$:

$$\hat{h}_{m+1}^{(0)} = \hat{h}_m^{(0)} + \delta_m f(\hat{h}_m^{(0)}, \hat{x}_m), \ \ \hat{h}_0^{(0)} = h_0, \tag{78}$$

$$\hat{h}_{m+1}^{(1)} = \hat{J}_m \hat{h}_m^{(1)} + \sqrt{\delta_m} \sigma(\hat{h}_m^{(0)}, \hat{x}_m) \xi_m, \ \ \hat{h}_0^{(1)} = 0, \tag{79}$$

$$\hat{h}_{m+1}^{(2)} = \hat{J}_m \hat{h}_m^{(2)} + \sqrt{\delta_m} + \delta_m \Psi_1(\hat{h}_m^{(0)}, \hat{h}_m^{(1)}) + \delta_m \Psi_2(\hat{h}_m^{(0)}, \hat{h}_m^{(1)}) \xi_m, \ \ \hat{h}_0^{(2)} = 0, \tag{80}$$

where the

$$\hat{J}_m = I + \delta_m f'(\hat{h}_m^{(0)}, \hat{x}_m) \tag{81}$$

are the state-to-state Jacobians and

$$\Psi_1(h_0, h_1) = \frac{1}{2} \sum_{i,j} \frac{\partial^2 f_m}{\partial h^i \partial h^j}(h_0) h_1^i h_1^j, \tag{82}$$

$$\Psi_2(h_0, h_1) = \sum_i \frac{\partial \sigma_m}{\partial h^i}(h_0) h_1^i. \tag{83}$$

Note that the above equations can also be obtained by E-M discretization of Eq. (60)-(61).

The following theorem is a discrete-time analogue of Theorem 7. Recall that the big $\mathcal{O}$ notation is to be understood in the almost sure sense.

**Theorem 9.** *Under the same assumption as before, for a scalar-valued loss function $\ell \in \mathcal{C}^2(\mathbb{R}^{d_h})$, for $m = 0, 1, \ldots, M - 1$, we have*

$$\ell(\hat{h}_{m+1}) = \ell(\hat{h}_m^{(0)}) + \epsilon \nabla \ell(\hat{h}_m^{(0)}) \cdot \hat{h}_m^{(1)} + \epsilon^2 \left( \nabla \ell(\hat{h}_m^{(0)}) \cdot \hat{h}_m^{(2)} + \frac{1}{2}(\hat{h}_m^{(1)})^\top \nabla^2 \ell(\hat{h}_m^{(0)})(\hat{h}_m^{(1)}) \right)$$
$$+ \mathcal{O}(\epsilon^3), \tag{84}$$

*as $\epsilon \to 0$, where the $\hat{h}_m^{(i)}$, $i = 0, 1, 2$, satisfy Eq.(78)-(80).*

*Proof.* The proof is analogous to the one for continuous-time case, working with the discrete-time process (77) instead of continuous-time process. $\qquad \square$

We begin by recalling a remark from the main text.

**Remark 1.** Interestingly, Theorem 8 looks like discrete-time analogue of Theorem 6 for CT-RNN, except that, unlike the term $Q$ there, the term $\hat{Q}$ for the discrete-time case has no explicit dependence on the *derivative (with respect to $h$) of the noise coefficient* $\sigma$. Therefore, a direct discretization of the result in Theorem 6 would not give us the correct explicit regularizer for discrete-time NRNNs. This remark highlights the difference between learning in the practical discrete-time setting versus learning in the idealized continuous-time setting with NRNNs. This also means that we need to work out an independently crafted proof for the discrete-time case.

The proof of Theorem 8 involves some tedious, albeit technically straightforward, computations. The key ingredients are the recursive relations (78)-(80) and the property of standard Gaussian random vectors that

$$\mathbb{E}\xi_p^l \xi_q^j = e_{pq} e_{lj}, \tag{85}$$

where the $e_{pq}$ denote the Kronecker delta.

To organize our proof, we begin by introducing some notation and proving a lemma.

**Notation.** For $m = 1, \ldots, M - 1$, let us denote $f'_m := f'(\hat{h}^{(0)}_m, \hat{x}_m)$, $\sigma_m := \sigma(\hat{h}^{(0)}_m, \hat{x}_m)$,

$$H_{lj} f^i_m := \frac{\partial^2 f^i(\hat{h}^{(0)}_m, \hat{x}_m)}{\partial[\hat{h}^{(0)}_m]^l \partial[\hat{h}^{(0)}_m]^j}, \tag{86}$$

$$D_l \sigma^{ij}_m := \frac{\partial \sigma^{ij}(\hat{h}^{(0)}_m, \hat{x}_m)}{\partial[\hat{h}^{(0)}_m]^l}, \tag{87}$$

and

$$\hat{\Phi}_{m,k} := J_m J_{m-1} \cdots J_k, \quad \hat{\Phi}_{k,k+1} = I, \tag{88}$$

for $k = 1, \ldots, m$. For computational convenience, we are using Einstein's summation notation for repeated indices in the following.

**Lemma 4.** *For $m = 0, 1, \ldots, M$, $\mathbb{E}\hat{h}^{(1)}_m = 0$ and*

$$\mathbb{E}[\hat{h}^{(1)}_m]^l[\hat{h}^{(1)}_m]^j = \delta_{m-1}\sigma^{ll_1}_{m-1}\sigma^{jl_1}_{m-1} + \sum_{k=1}^{m-1} \delta_{k-1}\hat{\Phi}^{ll_2}_{m-1,k}\hat{\Phi}^{jj_2}_{m-1,k}\sigma^{l_2 l_3}_{k-1}\sigma^{j_2 l_3}_{k-1}. \tag{89}$$

*Proof.* From Eq. (79), we have $\hat{h}^{(1)}_0 = 0$, $\hat{h}^{(1)}_1 = \sqrt{\delta_0}\sigma_{t_0}\xi_0$ and, upon iterating, for $m = 1, \ldots, M - 1$,

$$\hat{h}^{(1)}_{m+1} = \sqrt{\delta_m}\sigma_m\xi_m + \sum_{k=1}^{m} \sqrt{\delta_{k-1}}\hat{\Phi}_{m,k}\sigma_{k-1}\xi_{k-1}. \tag{90}$$

The first equality in the lemma follows from taking expectation of Eq. (90) and using the fact that the $\xi_k$ are (mean zero) standard Gaussian random variables. The second equality in the lemma follows from taking expectation of a product of components of the $\hat{h}^{(1)}_{m+1}$ in Eq. (90) and applying the property (85). $\qquad\square$

*Proof of Theorem 8.* Iterating Eq. (80), we obtain $\hat{h}^{(2)}_0 = 0$, $\hat{h}^{(2)}_1 = \delta_0\Psi_1(h_0, 0) + \sqrt{\delta_0}\Psi_2(h_0, 0)\xi_0$ and, for $m = 1, \ldots, M - 1$,

$$\hat{h}^{(2)}_{m+1} = \delta_m\Psi_1(\hat{h}^{(0)}_m, \hat{h}^{(1)}_m) + \sqrt{\delta_m}\Psi_2(\hat{h}^{(0)}_m, \hat{h}^{(1)}_m) + \sum_{k=1}^{m} \delta_{k-1}\hat{\Phi}_{m,k}\Psi_1(\hat{h}^{(0)}_{k-1}, \hat{h}^{(1)}_{k-1})$$

$$+ \sum_{k=1}^{m} \sqrt{\delta_{k-1}}\hat{\Phi}_{m,k}\Psi_1(\hat{h}^{(0)}_{k-1}, \hat{h}^{(1)}_{k-1})\xi_{k-1}. \tag{91}$$

Substituting in the formulae (82)-(83) in the right hand side above and then using Eq. (90):

$$[\hat{h}^{(2)}_{m+1}]^i = \frac{\delta_m}{2}[\hat{h}^{(1)}_m]^l H_{lj} f^i_m [\hat{h}^{(1)}_m]^j + \sum_{k=1}^{m} \frac{\delta_{k-1}}{2}\hat{\Phi}^{ip}_{m,k}[\hat{h}^{(1)}_{k-1}]^l H_{lj} f^p_{k-1} [\hat{h}^{(1)}_{k-1}]^j$$

$$+ \sqrt{\delta_m}D_l\sigma^{ij}_m[\hat{h}^{(1)}_m]^l\xi^j_m + \sum_{k=1}^{m} \sqrt{\delta_{k-1}}\hat{\Phi}^{iq}_{m,k}D_l\sigma^{qr}_{k-1}[\hat{h}^{(1)}_{k-1}]^l\xi^r_{k-1} \tag{92}$$

$$= \frac{\delta_m}{2}[\hat{h}^{(1)}_m]^l H_{lj} f^i_m [\hat{h}^{(1)}_m]^j + \sum_{k=1}^{m} \frac{\delta_{k-1}}{2}\hat{\Phi}^{ip}_{m,k}[\hat{h}^{(1)}_{k-1}]^l H_{lj} f^p_{k-1} [\hat{h}^{(1)}_{k-1}]^j$$

$$+ \sqrt{\delta_m}D_l\sigma^{ij}_m\xi^j_m \left( \sqrt{\delta_{m-1}}\sigma^{ll_1}_{m-1}\xi^{l_1}_{m-1} + \sum_{k=1}^{m-1} \sqrt{\delta_{k-1}}\hat{\Phi}^{ll_1}_{m-1,k}\sigma^{l_1 l_2}_{k-1}\xi^{l_2}_{k-1} \right)$$

$$+ \sqrt{\delta_1}\hat{\Phi}^{iq}_{m,2}D_l\sigma^{qr}_1\xi^r_1(\sqrt{\delta_0}\sigma^{ll_1}_0\xi^{l_1}_0) \tag{93}$$

$$+ \sum_{k=3}^{m} \sqrt{\delta_{k-1}}\hat{\Phi}^{iq}_{m,k}D_l\sigma^{qr}_{k-1}\xi^r_{k-1} \left( \sqrt{\delta_{k-2}}\sigma^{lp_1}_{k-2}\xi^{p_1}_{k-2} + \sum_{k'=1}^{k-2} \sqrt{\delta_{k'-1}}\hat{\Phi}^{lp_1}_{k-2,k'}\sigma^{p_1 p_2}_{k'-1}\xi^{p_2}_{k'-1} \right),$$

where we have made use of the fact that $\hat{h}^{(1)}_0 = 0$ and $\hat{h}^{(1)}_1 = \sqrt{\delta_0}\sigma_0\xi_0$ in the last two lines above to rewrite the summation (so that the summation over $k$ in the last line above starts at $k = 3$).

Therefore, using the above result, Lemma 4 and Eq. (85), we compute the expectation of $[\hat{h}_{m+1}^{(2)}]^i$:

$$\mathbb{E}[\hat{h}_{m+1}^{(2)}]^i = \frac{1}{2} \sum_{k=1}^{m+1} \delta_{k-1} \hat{\Phi}_{m,k}^{ip} H_{lj} f_m^p \sum_{k=1}^{m} \delta_{k-1} \hat{\Phi}_{m-1,k}^{ll_2} \sigma_{k-1}^{l_2 l_3} \sigma_{k-1}^{j_2 l_3} \hat{\Phi}_{m-1,k}^{jj_2}. \tag{94}$$

Moreover, using Lemma 4, we obtain, for $m = 1, 2, \ldots, M-1$,

$$\mathbb{E}[\hat{h}_{m+1}^{(1)}]^l [H_{\hat{h}^{(0)}} l]^{lj} [\hat{h}_{m+1}^{(1)}]^j = \sum_{k=1}^{m+1} \delta_{k-1} \sigma_{k-1}^{l_2 l_3} \hat{\Phi}_{m,k}^{ll_2} [H_{\hat{h}^{(0)}} l]^{lj} \hat{\Phi}_{m,k}^{jj_2} \sigma_{k-1}^{j_2 l_3}. \tag{95}$$

The first statement of the theorem then follows from Theorem 9 and Eq. (94)-(95) (with $m := M-1$):

$$\hat{Q}(\bar{h}^\delta) = \partial_i l(\bar{h}_M^\delta) \sum_{k=1}^{M} \delta_{k-1} \hat{\Phi}_{M-1,k}^{ip} \sum_{m=1}^{M-1} \delta_{m-1} \partial_{lj} [f_M]^p \hat{\Phi}_{M-2,m}^{ll_2} \sigma_{m-1}^{l_2 l_3} \sigma_{m-1}^{j_2 l_3} \hat{\Phi}_{M-2,m}^{jj_2}, \tag{96}$$

$$\hat{R}(\bar{h}^\delta) = \sum_{m=1}^{M} \delta_{m-1} \sigma_{m-1}^{l_2 l_3} \hat{\Phi}_{M-1,m}^{ll_2} [H_{\bar{h}^\delta} l]^{lj} \hat{\Phi}_{M-1,m}^{jj_2} \sigma_{m-1}^{j_2 l_3}. \tag{97}$$

The last statement of the theorem follows from taking straightforward bounds. $\qquad\square$

**Remark 2.** We remark that the computed $\hat{h}_m^{(2)}$ (a key step in the above proof), like that for $h_t^{(2)}$ in the continuous-time case, has explicit dependence on the noise coefficient. It is only upon taking the expectation (see Eq. (94)) that the dependence on the noise coefficient vanishes (whereas $\mathbb{E}h_t^{(2)} \neq 0$ retains its dependence on the noise coefficient). This fully reconciles with Remark 1.

**Remark 3.** Moreover, one can compute the variance of $l(\hat{h}_M)$ to be $\epsilon^2 (\nabla l(\hat{h}_M^{(0)}))^T C \nabla l(\hat{h}_M^{(0)}) + \mathcal{O}(\epsilon^3)$, as $\epsilon \to 0$, where $C$ is a PSD matrix whose $(l, j)$-entry is given by Eq. (89) with $m := M$. So we see that the spread of $l(\hat{h}_M)$ about its average is $\mathcal{O}(\epsilon^2)$ as $\epsilon \to 0$.

# E  Bound on Classification Margin and a Generalization Bound for Deterministic RNNs: Proof of Theorem 2 in the Main Paper

We recall the setting considered in the main paper before providing proof to the results presented there.

Let $\mathcal{S}_N$ denote a set of training samples $s_n := (\boldsymbol{x}_n, y_n)$ for $n = 1, \ldots, N$, where each input sequence $\boldsymbol{x}_n = (x_{n,0}, x_{n,1}, \ldots, x_{n,M-1}) \in \mathcal{X} \subset \mathbb{R}^{d_x M}$ has a corresponding class label $y_n \in \mathcal{Y} = \{1, \ldots, d_y\}$. Following the statistical learning framework, these samples are assumed to be independently drawn from an underlying probability distribution $\mu$ on the sample space $\mathcal{S} = \mathcal{X} \times \mathcal{Y}$. An RNN-based classifier $g^\delta(\boldsymbol{x})$ is constructed in the usual way by taking

$$g^\delta(\boldsymbol{x}) = \text{argmax}_{i=1,\ldots,d_y} p^i(V \bar{h}_M^\delta[\boldsymbol{x}]), \tag{98}$$

where $p^i(x) = e^{x^i} / \sum_j e^{x^j}$ is the softmax function. Letting $\ell$ denoting the cross-entropy loss, such a classifier is trained from $\mathcal{S}_N$ by minimizing the empirical risk (training error)

$$\mathcal{R}_N(g^\delta) := \frac{1}{N} \sum_{n=1}^{N} \ell(g^\delta(\boldsymbol{x}_n), y_n)$$

as a proxy for the true (population) risk (testing error) $\mathcal{R}(g^\delta) = \mathbb{E}_{(\boldsymbol{x},y)\sim\mu} \ell(g^\delta(\boldsymbol{x}), y)$ with $(\boldsymbol{x}, y) \in \mathcal{S}$.

The measure used to quantify the prediction quality is the *generalization error* (or estimation error), which is the difference between the empirical risk of the classifier on the training set and the true risk:

$$GE(g^\delta) := |\mathcal{R}(g^\delta) - \mathcal{R}_N(g^\delta)|. \tag{99}$$

The classifier is a function of the output of the deterministic RNN, which is an Euler discretization of the ODE (1) in the main paper with step sizes $\delta = (\delta_m)$. In particular, for the Lipschitz RNN,

$$\hat{\Phi}_{m,k} = \hat{J}_m \hat{J}_{m-1} \cdots \hat{J}_k, \tag{100}$$

where $\hat{J}_l = I + \delta_l(A + D_l W)$, with $D_l^{ij} = a'([W\bar{h}_l^\delta + U\hat{x}_l + b]^i)e_{ij}$.

In the following, we let $\mathrm{conv}(\mathcal{X})$ denote the convex hull of $\mathcal{X}$. We denote $\hat{\boldsymbol{x}}_{0:m} := (\hat{x}_0, \ldots, \hat{x}_m)$ so that $\hat{\boldsymbol{x}} = \hat{\boldsymbol{x}}_{0:M-1}$, and use the notation $f[\boldsymbol{x}]$ to indicate the dependence of the function $f$ on the vector $\boldsymbol{x}$. Moreover, we will need the following two definitions to characterize a training sample $s_i = (\boldsymbol{x}_i, y_i)$

Working in the above setting, we now recall and prove the second main result in the main paper, providing bounds for classification margin for the deterministic RNN classifiers $g^\delta$.

**Theorem 10** (Classification margin bound for the deterministic RNN – Theorem 2 in the main paper). *Suppose that Assumption A in the main paper holds. Assume that the $o(s_i) > 0$ and*

$$\gamma(s_i) := \frac{o(s_i)}{C \sum_{m=0}^{M-1} \delta_m \sup_{\hat{\boldsymbol{x}} \in \mathrm{conv}(\mathcal{X})} \|\hat{\Phi}_{M,m+1}[\hat{\boldsymbol{x}}]\|_2} > 0, \tag{101}$$

*where*

$$C = \|V\|_2 \left( \max_{m=0,1,\ldots,M-1} \left\| \frac{\partial f(\bar{h}_m^\delta, \hat{x}_m)}{\partial \hat{x}_m} \right\|_2 \right) > 0$$

*is a constant (in particular, $C = \|V\|_2 \left(\max_{m=0,\ldots,M-1} \|D_m U\|_2\right)$ for Lipschitz RNNs), the $\hat{\Phi}_{m,k}$ are defined in* (100) *and the $\delta_m$ are the step sizes. Then, we have the following upper bound on the classification margin for the training sample $s_i$:*

$$\gamma^d(s_i) \geq \gamma(s_i). \tag{102}$$

Moreover, under additional assumptions one can obtain the following generalization bound, which follows from Theorem 10.

**Theorem 11** (A generalization bound for the deterministic RNN). *Under the same setting as Theorem 10, if we further assume that $\mathcal{X}$ is a (subset of) $k$-dimensional manifold with $k \leq d_x M$, $\gamma := \min_{s_i \in \mathcal{S}_N} \gamma(s_i) > 0$, and $\ell(g^\delta(\boldsymbol{x}), y) \leq L_g$ for all $s \in \mathcal{S}$, then for any $\delta' > 0$, with probability at least $1 - \delta'$,*

$$GE(g^\delta) \leq L_g \left( \frac{1}{\gamma^{k/2}} \sqrt{\frac{d_y C_M^k 2^{k+1} \log 2}{N}} + \sqrt{\frac{2 \log(1/\delta')}{N}} \right), \tag{103}$$

*where $C_M > 0$ is a constant that measures complexity of $\mathcal{X}$, $N$ is the number of training examples and $d_y$ is the number of label classes.*

**Remark 4.** Generalization bounds involving classification margins (for RNNs in particular) are a separate topic with a significant presence in the literature. We emphasize that the generalization bound above is one of the many bounds that one can derive for RNNs. There exist much tighter bounds (for various variants of RNNs under various assumptions and settings) which may be equally applicable and lead to the same claimed conclusion, but are much more difficult to state (see, for instance, Theorem E.1 in [28]). There are also other types of generalization bounds that are not obtained in terms of classification margin in the literature. Although they are interesting in their own, our focus here is on bounds that can be expressed in terms of classification margin. Therefore, meaningful comparisons between these generalization bounds are not straightforward.

In order to prove Theorem 10 and Theorem 11, we place ourselves in the algorithmic robustness framework of [30]. This framework provides bounds for the generalization error based on the robustness of a learning algorithm that learns a classifier $g$ by exploiting the structure of the training set $\mathcal{S}_N$. Robustness is, roughly speaking, the desirable property for a learning algorithm that if a testing sample is "similar" to a training sample, then the testing error is close to the training error (i.e., the algorithm is insensitive to small perturbations in the training data).

To ensure that our exposition is self-contained, we recall important definitions and results from [30, 27] to formalize the previous statement in the context of our deterministic RNNs in the following.

**Definition 4.** Let $\mathcal{S}_N$ be a training set and $\mathcal{S}$ the sample space. A learning algorithm is $(K, \epsilon(\mathcal{S}_N))$-robust if $\mathcal{S}$ can be partitioned into $K$ disjoint sets denoted by $\mathcal{K}_k$, $k = 1, \ldots, K$:

$$\mathcal{K}_k \subset \mathcal{S}, \ k = 1, \ldots, K, \tag{104}$$

$$\mathcal{S} = \cup_{k=1}^{K} \mathcal{K}_k, \ \text{ and } \ \mathcal{K}_k \cap \mathcal{K}_{k'} = \emptyset, \forall k \neq k', \tag{105}$$

such that for all $s_i \in \mathcal{S}_N$ and all $s \in \mathcal{S}$,

$$s_i = (\boldsymbol{x}_i, y_i) \in \mathcal{K}_k \wedge s = (\boldsymbol{x}, y) \in \mathcal{K}_k \implies |\ell(g(\boldsymbol{x}_i), y_i) - \ell(g(\boldsymbol{x}), y)| \leq \epsilon(\mathcal{S}_N). \tag{106}$$

The above definition says that a robust learning algorithm selects a classifier $g$ for which the losses of any $s$ and $s_i$ in the same partition $\mathcal{K}_k$ are close.

The following result from Theorem 1 in [30] will be critical to the proof of Theorem 10. It provides a generalization bound for robust algorithms.

**Theorem 12.** *If a learning algorithm is $(K, \epsilon(\mathcal{S}_N))$-robust and $\ell(g(\boldsymbol{x}), y) \leq M$ for all $s = (\boldsymbol{x}, y) \in \mathcal{S}$, for some constant $M > 0$, then for any $\delta > 0$, with probability at least $1 - \delta$,*

$$GE(g) \leq \epsilon(\mathcal{S}_N) + M\sqrt{\frac{2K \log(2) + 2\log(1/\delta)}{m}}. \tag{107}$$

Note that the above generalization bound is data-dependent, in contrast to bounds obtained via approaches based on complexity or stability arguments that give bounds in terms of data agnostic measures such as the Rademacher complexity or the VC dimension, which are found not sufficient for explaining the good generalization properties of deep neural networks.

The number of partition $K$ in the above can be bounded in terms of the covering number of the sample space $\mathcal{S}$, which gives a way to measure the complexity of sets. We recall the definition of covering number in the following.

**Definition 5** (Covering). Let $\mathcal{A}$ be a set. We say that $\mathcal{A}$ is $\rho$-covered by a set $\mathcal{A}'$, with respect to the (pseudo-)metric $d$, if for all $a \in \mathcal{A}$, there exists $a' \in \mathcal{A}'$ with $d(a, a') \leq \rho$. We call the cardinality of the smallest $\mathcal{A}'$ that $\rho$-covers $\mathcal{A}$ covering number, denoted by $\mathcal{N}(\mathcal{S}; d, \rho)$.

The covering number is the smallest number of (pseudo-)metric balls of radius $\rho$ needed to cover $\mathcal{S}$ and we denote it by $\mathcal{N}(\mathcal{S}; d, \rho)$, where $d$ denotes the (pseudo-)metric. The choice of metric $d$ determines how efficiently one may cover $\mathcal{X}$. For example, the Euclidean metric $d(x, x') = \|x - x'\|_2$ for $x, x' \in \mathcal{X}$. The covering number of many structured low-dimensional data models can be bounded in terms of their intrinsic properties. Since in our case the space $\mathcal{S} = \mathcal{X} \times \mathcal{Y}$, we write $\mathcal{N}(\mathcal{S}; d, \rho) \leq d_y \cdot \mathcal{N}(\mathcal{X}; d, \rho)$, where $d_y$ is the number of label classes. We take $d$ to be the Euclidean metric: $d(\boldsymbol{x}, \boldsymbol{x}') = \|\boldsymbol{x} - \boldsymbol{x}'\|_2$ for $\boldsymbol{x}, \boldsymbol{x}' \in \mathcal{X}$, unless stated otherwise.

**Lemma 5** (Example 27.1 from [26]). *Assume that $\mathcal{X} \subset \mathbb{R}^m$ lies in a $k$-dimensional subspace of $\mathbb{R}^m$. Let $c = \max_{x \in \mathcal{X}} \|x\|$ and take $d$ to be the Euclidean metric. Then $\mathcal{N}(\mathcal{X}; d, \rho) \leq (2c\sqrt{k}/\rho)^k$.*

In other words, a subset, $\mathcal{X}$, of a $k$-dimensional manifold has the covering number $(C_M/\rho)^k$, where $C_M > 0$ is a constant. We remark that other complexity measures such as Rademacher complexity can be bounded based on the covering number (see [26] for details).

The class of robust learning algorithms that is of interest to us is the large margin classifiers. We define classification margin in the following.

**Definition 6** (Classification margin). The classification margin of a training sample $s_i = (\boldsymbol{x}_i, y_i)$ measured by a metric $d$ is defined as the radius of the largest $d$-metric ball in $\mathcal{X}$ centered at $\boldsymbol{x}_i$ that is contained in the decision region associated with the class label $y_i$, i.e., it is:

$$\gamma^d(s_i) = \sup\{a : d(\boldsymbol{x}_i, \boldsymbol{x}) \leq a \implies g(\boldsymbol{x}) = y_i \ \forall \boldsymbol{x}\}. \tag{108}$$

Intuitively, a larger classification margin allows a classifier to associate a larger region centered on a point $\boldsymbol{x}_i$ in the input space to the same class. This makes the classifier less sensitive to input perturbations and a noisy perturbation of $\boldsymbol{x}_i$ is still likely to fall within this region, keeping the classifier prediction. In this sense, the classifier becomes more robust.

The following result follows from Example 9 in [30].

**Proposition 2.** *If there exists a $\gamma > 0$ such that $\gamma^d(s_i) > \gamma$ for all $s_i \in \mathcal{S}_N$, then the classifier $g$ is $(d_y \cdot \mathcal{N}(\mathcal{X}; d, \gamma/2), 0)$-robust.*

In our case the networks are trained by a loss (cross-entropy) that promotes separation of different classes at the network output. The training aims at maximizing a certain notion of score of each training sample.

**Definition 7** (Score). For a training sample $s_i = (\boldsymbol{x}_i, y_i)$, we define its score as

$$o(s_i) = \min_{j \neq y_i} \sqrt{2}(e_{y_i} - e_j)^T S^\delta[\boldsymbol{x}_i] \geq 0, \tag{109}$$

where $e_i \in \mathbb{R}^{d_y}$ is the Kronecker delta vector with $e_i^i = 1$ and $e_i^j = 0$ for $i \neq j$, $S^\delta[\boldsymbol{x}_i] := p(V\bar{h}_M^\delta[\boldsymbol{x}_i])$ with $\bar{h}_M^\delta[\boldsymbol{x}_i]$ denoting the hidden state of the RNN, driven by the input sequence $\boldsymbol{x}_i$, at terminal index $M$.

The RNN classifier $g^\delta$ is defined as

$$g^\delta(\boldsymbol{x}) = \arg \max_{i \in \{1, \ldots, d_y\}} S^i[\boldsymbol{x}], \tag{110}$$

and the decision boundary between class $i$ and class $j$ in the output space is given by the hyperplane $\{z = p(V\bar{h}_M^\delta) : z^i = z^j\}$. A positive score implies that at the network output, classes are separated by a margin that corresponds to the score. However, a large score may not imply a large classification margin – recall that the classification margin is a function of the decision boundary in the input space, whereas the training algorithm aims at optimizing the decision boundary at the network output in the output space.

We need the following lemma relating a pair of vectors in the input space and the output space.

**Lemma 6.** *For any $\boldsymbol{x}, \boldsymbol{x}' \in \mathcal{X} \subset \mathbb{R}^{d_x M}$, and a given RNN output functional $\mathcal{F}[\cdot]$,*

$$\|\mathcal{F}[\boldsymbol{x}] - \mathcal{F}[\boldsymbol{x}']\|_2 \leq \sup_{\bar{\boldsymbol{x}} \in \operatorname{conv}(\mathcal{X})} \|J[\bar{\boldsymbol{x}}]\|_2 \cdot \|\boldsymbol{x} - \boldsymbol{x}'\|_2, \tag{111}$$

*where $J[\boldsymbol{x}] = d\mathcal{F}[\boldsymbol{x}]/d\boldsymbol{x}$ is the input-output Jacobian of the RNN output functional.*

*Proof.* Let $t \in [0, 1]$ and define the function $F(t) = \mathcal{F}[\boldsymbol{x} + t(\boldsymbol{x}' - \boldsymbol{x})]$. Note that

$$\frac{dF(t)}{dt} = J[\boldsymbol{x} + t(\boldsymbol{x}' - \boldsymbol{x})](\boldsymbol{x}' - \boldsymbol{x}). \tag{112}$$

Therefore,

$$\mathcal{F}[\boldsymbol{x}'] - \mathcal{F}[\boldsymbol{x}] = F(1) - F(0) = \int_0^1 \frac{dF(t)}{dt} dt = \left(\int_0^1 J[\boldsymbol{x} + t(\boldsymbol{x}' - \boldsymbol{x})]dt\right)(\boldsymbol{x}' - \boldsymbol{x}), \tag{113}$$

where we have used the fundamental theorem of calculus.

Now,

$$\|\mathcal{F}[\boldsymbol{x}] - \mathcal{F}[\boldsymbol{x}']\|_2 \leq \left\|\int_0^1 J[\boldsymbol{x} + t(\boldsymbol{x}' - \boldsymbol{x})]dt\right\|_2 \cdot \|(\boldsymbol{x}' - \boldsymbol{x})\|_2 \tag{114}$$

$$\leq \sup_{\boldsymbol{x}, \boldsymbol{x}' \in \mathcal{X}, t \in [0,1]} \|J[\boldsymbol{x} + t(\boldsymbol{x}' - \boldsymbol{x})]\|_2 \cdot \|(\boldsymbol{x}' - \boldsymbol{x})\|_2 \tag{115}$$

$$\leq \sup_{\bar{\boldsymbol{x}} \in \operatorname{conv}(\mathcal{X})} \|J[\bar{\boldsymbol{x}}]\|_2 \cdot \|\boldsymbol{x} - \boldsymbol{x}'\|_2, \tag{116}$$

where we have used the fact that $\boldsymbol{x} + t(\boldsymbol{x}' - \boldsymbol{x}) \in \operatorname{conv}(\mathcal{X})$ for all $t \in [0, 1]$ to arrive at the last line. The proof is done. $\square$

The classification margin depends on the score and the network's expansion and contraction of distances around the training points. These can be quantified by studying the network's input-output Jacobian matrix. The following proposition provides classification margin bounds in terms of the score and input-output Jacobian associated to the RNN classifier.

**Proposition 3.** *Assume that a RNN classifier $g^\delta(\boldsymbol{x})$, defined in (110), classifies a training sample $\boldsymbol{x}_i$ with the score $o(s_i) > 0$. Then we have the following lower bound for the classification margin:*

$$\gamma^d(s_i) \geq \frac{o(s_i)}{\sup_{\boldsymbol{x} \in \mathrm{conv}(\mathcal{X})} \|J[\boldsymbol{x}]\|_2}, \tag{117}$$

*where $\mathrm{conv}(\mathcal{X})$ denotes the convex hull of $\mathcal{X}$ and $J[\boldsymbol{x}] = d\mathcal{F}[\boldsymbol{x}]/d\boldsymbol{x}$, with $\mathcal{F}[\boldsymbol{x}] = p(V\bar{h}_M^\delta[\boldsymbol{x}])$, is the input-output Jacobian associated to the RNN.*

*Proof.* The proof is essentially identical to that of Theorem 4 in [27]. We provide the full detail here for completeness.

Denote $o(s_i) = o(\boldsymbol{x}^{(i)}, y^{(i)})$, where $\boldsymbol{x}^{(i)} := (x_0^{(i)}, \cdots, x_{M-1}^{(i)}) \in \mathcal{X} \subset \mathbb{R}^{d_x M}$, and $v_{ij} = \sqrt{2}(e_i - e_j)$, where $e_i \in \mathbb{R}^{d_y}$ denotes the Kronecker delta vector.

The classification margin of the training sample $s_i$ is:

$$\gamma^d(s_i) = \sup\{a : \|\boldsymbol{x}^{(i)} - \boldsymbol{x}\|_2 \leq a \implies g^\delta(\boldsymbol{x}) = y^{(i)} \ \forall \boldsymbol{x}\} \tag{118}$$

$$= \sup\{a : \|\boldsymbol{x}^{(i)} - \boldsymbol{x}\|_2 \leq a \implies o(\boldsymbol{x}, y^{(i)}) > 0 \ \forall \boldsymbol{x}\}. \tag{119}$$

By Definition 7, $o(\boldsymbol{x}, y^{(i)}) > 0$ if and only if $\min_{j \neq y^{(i)}} v_{y^{(i)}j}^T \mathcal{F}[\boldsymbol{x}] > 0$.

On the other hand,

$$\min_{j \neq y^{(i)}} v_{y^{(i)}j}^T \mathcal{F}[\boldsymbol{x}] = \min_{j \neq y^{(i)}} (v_{y^{(i)}j}^T \mathcal{F}[\boldsymbol{x}^{(i)}] + v_{y^{(i)}j}^T (\mathcal{F}[\boldsymbol{x}] - \mathcal{F}[\boldsymbol{x}^{(i)}])) \tag{120}$$

$$\geq \min_{j \neq y^{(i)}} v_{y^{(i)}j}^T \mathcal{F}[\boldsymbol{x}^{(i)}] + \min_{j \neq y^{(i)}} v_{y^{(i)}j}^T (\mathcal{F}[\boldsymbol{x}] - \mathcal{F}[\boldsymbol{x}^{(i)}]) \tag{121}$$

$$= o(\boldsymbol{x}^{(i)}, y^{(i)}) + \min_{j \neq y^{(i)}} v_{y^{(i)}j}^T (\mathcal{F}[\boldsymbol{x}] - \mathcal{F}[\boldsymbol{x}^{(i)}]). \tag{122}$$

Therefore, $o(\boldsymbol{x}^{(i)}, y^{(i)}) + \min_{j \neq y^{(i)}} v_{y^{(i)}j}^T (\mathcal{F}[\boldsymbol{x}] - \mathcal{F}[\boldsymbol{x}^{(i)}]) > 0$ implies that $o(\boldsymbol{x}, y^{(i)}) > 0$ and so

$$\gamma^d(s_i) \geq \sup \left\{ a : \|\boldsymbol{x}^{(i)} - \boldsymbol{x}\|_2 \leq a \implies o(\boldsymbol{x}^{(i)}, y^{(i)}) + \min_{j \neq y^{(i)}} v_{y^{(i)}j}^T (\mathcal{F}[\boldsymbol{x}] - \mathcal{F}[\boldsymbol{x}^{(i)}]) > 0 \ \forall \boldsymbol{x} \right\} \tag{123}$$

$$= \sup \left\{ a : \|\boldsymbol{x}^{(i)} - \boldsymbol{x}\|_2 \leq a \implies o(\boldsymbol{x}^{(i)}, y^{(i)}) - \max_{j \neq y^{(i)}} v_{y^{(i)}j}^T (\mathcal{F}[\boldsymbol{x}^{(i)}] - \mathcal{F}[\boldsymbol{x}]) > 0 \ \forall \boldsymbol{x} \right\} \tag{124}$$

$$= \sup \left\{ a : \|\boldsymbol{x}^{(i)} - \boldsymbol{x}\|_2 \leq a \implies o(\boldsymbol{x}^{(i)}, y^{(i)}) > \max_{j \neq y^{(i)}} v_{y^{(i)}j}^T (\mathcal{F}[\boldsymbol{x}^{(i)}] - \mathcal{F}[\boldsymbol{x}]) \ \forall \boldsymbol{x} \right\}. \tag{125}$$

Now, using the fact that $\|v_{y^{(i)}j}\|_2 = 1$ and Lemma 6, we have:

$$\max_{j \neq y^{(i)}} v_{y^{(i)}j}^T (\mathcal{F}[\boldsymbol{x}^{(i)}] - \mathcal{F}[\boldsymbol{x}]) \leq \sup_{\bar{\boldsymbol{x}} \in \mathrm{conv}(\mathcal{X})} \|J[\bar{\boldsymbol{x}}]\|_2 \cdot \|\boldsymbol{x}^{(i)} - \boldsymbol{x}\|_2. \tag{126}$$

Using this inequality gives:

$$\gamma^d(s_i) \geq \sup \left\{ a : \|\boldsymbol{x}^{(i)} - \boldsymbol{x}\|_2 \leq a \implies o(\boldsymbol{x}^{(i)}, y^{(i)}) > \sup_{\bar{\boldsymbol{x}} \in \mathrm{conv}(\mathcal{X})} \|J[\bar{\boldsymbol{x}}]\|_2 \cdot \|\boldsymbol{x}^{(i)} - \boldsymbol{x}\|_2 \ \forall \boldsymbol{x} \right\} \tag{127}$$

$$\geq \frac{o(\boldsymbol{x}^{(i)}, y^{(i)})}{\sup_{\bar{\boldsymbol{x}} \in \mathrm{conv}(\mathcal{X})} \|J[\bar{\boldsymbol{x}}]\|_2}. \tag{128}$$

The proof is done. $\qquad\qquad\square$

We now have all the needed ingredients to prove Theorem 10 and Theorem 11.

*Proof of Theorem 10 and Theorem 11.* By Proposition 2, Lemma 5, and our assumption on complexity of the sample space, the RNN classifier is $(d_y \cdot (2C_M/\gamma)^k, 0)$-robust, for some constant $C_M > 0$. Due to Theorem 12 (with $M := L_g$ there), it remains to prove the upper bound (102) for the classification margin of a training sample to complete the proof. Theorem 11 then follows from Theorem 12 (with $M := L_g$ there) and the inequality (103) follows immediately from Theorem 12.

By Proposition 3, we have

$$\gamma^d(s_i) \geq \frac{o(s_i)}{\sup_{\hat{\boldsymbol{x}} \in \text{conv}(\mathcal{X})} \|J[\hat{\boldsymbol{x}}]\|_2}, \tag{129}$$

where $J[\hat{\boldsymbol{x}}] := d\mathcal{F}[\hat{\boldsymbol{x}}]/d\hat{\boldsymbol{x}}$ is the input-output Jacobian associated to the RNN. Therefore, to complete the proof it suffices to show that

$$\|J[\hat{\boldsymbol{x}}]\|_2 \leq C \sum_{m=0}^{M-1} \delta_m \|\hat{\Phi}_{M,m+1}[\hat{\boldsymbol{x}}]\|_2, \tag{130}$$

where $C$ is the constant from the theorem and $\hat{\Phi}_{m+1,k}$, $0 \leq k \leq m \leq M-1$ satisfies:

$$\hat{\Phi}_{k,k} = I, \tag{131}$$

$$\hat{\Phi}_{m+1,k} = \hat{J}_m \hat{\Phi}_{m,k}, \tag{132}$$

where $\hat{J}_m = I + \delta_m f'(\hat{h}_m^0, \hat{x}_m)$ (with the $\hat{h}_m^{(0)}$ satisfying Eq. (78), recalling that we are replacing the superscript $\delta$ by hat when denoting the $\delta$-dependent approximating solutions for the sake of notation cleanliness) and the $\delta_m > 0$ are the step sizes.

Iterating (132) up to the $(m+1)$th step, for $m \geq k$, gives:

$$\hat{\Phi}_{m+1,k} = \hat{J}_m \hat{J}_{m-1} \cdots \hat{J}_k =: \prod_{l=k}^{m} \hat{J}_l. \tag{133}$$

Note that

$$\hat{\Phi}_{m+1,k} = \frac{\partial \hat{h}_{m+1}^{(0)}}{\partial \hat{h}_m^{(0)}} \frac{\partial \hat{h}_m^{(0)}}{\partial \hat{h}_{m-1}^{(0)}} \cdots \frac{\partial \hat{h}_{k+1}^{(0)}}{\partial \hat{h}_k^{(0)}} = \frac{d\hat{h}_{m+1}^{(0)}}{d\hat{h}_k^{(0)}}. \tag{134}$$

Now, applying chain rule:

$$J[\hat{\boldsymbol{x}}] = \frac{\partial p(V\hat{h}_M^{(0)})}{\partial \hat{h}_M^{(0)}} \sum_{j=0}^{M-1} \frac{\partial \hat{h}_M^{(0)}}{\partial \hat{h}_{M-1}^{(0)}} \cdots \frac{\partial \hat{h}_{j+2}^{(0)}}{\partial \hat{h}_{j+1}^{(0)}} \frac{\partial \hat{h}_{j+1}^{(0)}}{\partial \hat{x}_j}, \tag{135}$$

where $p$ is the softmax function.

We compute:

$$\frac{\partial p(V\hat{h}_M^{(0)})}{\partial \hat{h}_M^{(0)}} = VE, \tag{136}$$

where $E^{ij} = p^i(e^{ij} - p^j)$. From (134), we have

$$\frac{\partial \hat{h}_M^{(0)}}{\partial \hat{h}_{M-1}^{(0)}} \cdots \frac{\partial \hat{h}_{j+2}^{(0)}}{\partial \hat{h}_{j+1}^{(0)}} = \hat{\Phi}_{M,j+1}. \tag{137}$$

On the other hand,

$$\frac{\partial \hat{h}_{j+1}^{(0)}}{\partial \hat{x}_j} = \delta_j \frac{\partial f(\hat{h}_j^{(0)}, \hat{x}_j)}{\partial \hat{x}_j}, \tag{138}$$

for $j = 0, 1, \ldots, M-1$. Note that for Lipschitz RNNs, we have $\frac{\partial \hat{h}_{j+1}^{(0)}}{\partial \hat{x}_j} = \delta_j D_j U$, where $D_l^{ij} = a'([W\hat{h}_l^{(0)} + U\hat{x}_l + b]^i)e_{ij}$.

Using the results of the above computations gives:

$$J[\hat{\boldsymbol{x}}] = VE \sum_{m=0}^{M-1} \delta_m \hat{\Phi}_{M,m+1} \frac{\partial f(\hat{h}_m^{(0)}, \hat{x}_m)}{\partial \hat{x}_m}. \tag{139}$$

Therefore,

$$\|J[\hat{\boldsymbol{x}}]\|_2 \le \|VE\|_2 \sum_{m=0}^{M-1} \|\delta_m \hat{\Phi}_{M,m+1}\|_2 \left\| \frac{\partial f(\hat{h}_m^{(0)}, \hat{x}_m)}{\partial \hat{x}_m} \right\|_2 \tag{140}$$

$$\le \|V\|_2 \left( \max_{m=0,1,\ldots,M-1} \left\| \frac{\partial f(\hat{h}_m^{(0)}, \hat{x}_m)}{\partial \hat{x}_m} \right\|_2 \right) \sum_{m=0}^{M-1} \delta_m \|\hat{\Phi}_{M,m+1}\|_2 \tag{141}$$

$$= C \sum_{m=0}^{M-1} \delta_m \|\hat{\Phi}_{M,m+1}\|_2. \tag{142}$$

For the Lipschitz RNN, we have $C := \|V\|(max_{m=0,1,\ldots,M-1} \|D_m U\|_2)$. The proof is done. $\quad\square$

It follows immediately from Eq. (142) that we have the following sufficient condition for stability with respect to hidden states of deterministic RNN to guarantee stability with respect to input sequence.

**Corollary 1.** *Fix a $M$ and assume that $C \sum_{m=0}^{M-1} \delta_m < 1$. Then, $\|\hat{\Phi}_{M,m+1}\|_2 \le 1$ for $m = 0, \ldots, M-1$ implies that $\|J[\hat{\boldsymbol{x}}]\|_2 < 1$.*

## F Stability and Noise-Induced Stabilization for NRNNs: Proof of Theorem 3 in the Main Paper

We begin by discussing stochastic stability for SDEs, which are the underlying continuous-time models for our NRNNs.

Although the additional complexities of SDEs over ODEs often necessitate more involved analyses, many of the same ideas typically carry across. This is also true for stability. A typical approach for proving stability of ODEs involves Lyapunov functions — in Chapter 4 of [23], such approaches are extended for SDEs. This gives way to three notions of stability: (1) stability in probability; (2) moment stability; and (3) almost sure stability. Their definitions are provided in Definitions 4.2.1, 4.3.1, 4.4.1 in [23], and are repeated below for convenience.

To preface the definition, consider initializing (23) at two different random variables $h_0$ and $h_0' := h_0 + \epsilon_0$, where $\epsilon_0 \in \mathbb{R}^{d_h}$ is a constant non-random perturbation with $\|\epsilon_0\| \le \delta$. The resulting hidden states, $h_t$ and $h_t'$, are set to satisfy (23) with the same Brownian motion $B_t$, starting from their initial values $h_0$ and $h_0'$, respectively. The evolution of $\epsilon_t = h_t' - h_t$ satisfies

$$\mathrm{d}\epsilon_t = A\epsilon_t\mathrm{d}t + \Delta a_t(\epsilon_t)\mathrm{d}t + \Delta\sigma_t(\epsilon_t)\mathrm{d}B_t, \tag{143}$$

where $\Delta a_t(\epsilon_t) = a(Wh_t' + Ux_t + b) - a(Wh_t + Ux_t + b)$ and $\Delta\sigma_t(\epsilon_t) = \sigma_t(h_t + \epsilon_t) - \sigma_t(h_t)$. Since $\Delta a_t(0) = 0$, $\Delta\sigma_t(0) = 0$ for all $t \in [0, T]$, $\epsilon_t = 0$ admits a trivial *equilibrium* for (143).

**Definition 8** (Stability for SDEs). The trivial solution of the SDE (143) is

    (i) *stochastically stable* (or, stable in probability) if for every $\epsilon \in (0, 1)$, $r > 0$, there exists a $\delta = \delta(\epsilon, r) > 0$ such that $\mathbb{P}(\|\epsilon_t\| < r \text{ for all } t \ge 0) \ge 1 - \epsilon$ whenever $\|\epsilon_0\| < \delta$.

    (ii) *stochastically asymptotically stable* if it is stochastically stable and, moreover, for every $\epsilon \in (0, 1)$, there exists a $\delta_0 = \delta_0(\epsilon) > 0$ such that $\mathbb{P}(\lim_{t\to\infty} \epsilon_t = 0) \ge 1 - \epsilon$ whenever $\|\epsilon_0\| < \delta_0$.

    (iii) *almost surely exponentially stable* if $\limsup_{t\to\infty} t^{-1} \log\|\epsilon_t\| < 0$ with probability one whenever $\|\epsilon_0\| < \delta_1$.

    (iv) *$p$-th moment exponentially stable* if there exists $\lambda, C > 0$ such that $\mathbb{E}\|\epsilon_t\|^p \le C\|\epsilon_0\|^p e^{-\lambda(t-t_0)}$ for all $t \ge t_0$.

The properties in Definition 8 are said to hold globally if they also hold under no restrictions on $\epsilon_0$. Stability in probability neglects to quantify rates of convergence, and is implied by almost sure exponential stability. On the other hand, for our class of SDEs, $p$-th moment exponential stability would imply almost sure exponential stability (see Theorem 4.2 in [23]).

One critical difference between Lyapunov stability theory for ODEs and SDEs lies in the *stochastic stabilization phenomenon*. Let $L$ be the infinitesimal generator (for a given input signal $x_t$) of the diffusion process described by the SDE (143):

$$L = \frac{\partial}{\partial t} + \sum_i \left( (A\epsilon)^i + \Delta a_t^i(\epsilon) \right) \frac{\partial}{\partial \epsilon^i} + \frac{1}{2} \sum_{i,j} \left[ \Delta \sigma_t(\epsilon) \Delta \sigma_t(\epsilon)^\top \right]^{ij} \frac{\partial^2}{\partial \epsilon^i \partial \epsilon^j}. \tag{144}$$

The generator for the corresponding ODE arises by taking $\Delta \sigma_t \equiv 0$. In classical Lyapunov theory for ODEs, the existence of a non-negative Lyapunov function $V$ satisfying $LV \leq 0$ in some neighbourhood of the equilibrium is both necessary and sufficient for stability (see Chapter 4 of [15]). For SDEs, it has been shown that this condition is sufficient, but no longer necessary [23, 22]. This is by the nature of stochastic stabilization — the addition of noise can can have the surprising effect of *increased* stability over its deterministic counterpart. Of course, this is not universally the case as some forms of noise can be sufficiently extreme to induce instability; see Section 4.5 in [23].

Identifying sufficient conditions which quantify the stochastic stabilization phenomenon are especially useful in our setting, and as it turns out (see also [19]), these are most easily obtained for almost sure exponential stability. Therefore, our stability analysis will focus on establishing *almost sure exponential stability*. The objective is to analyze such stability of the solution $\epsilon_t = 0$, that is, to see how the final state $\epsilon_T$ (and hence the output $y_T' - y_T = V\epsilon_T$ of the RNN) changes for an arbitrarily small initial perturbation $\epsilon_0 \neq 0$.

To this end, we consider an extension of the Lyapunov exponent to SDEs at the level of sample path [23].

**Definition 9** (Almost sure global exponential stability). The sample (or pathwise) Lyapunov exponent of the trivial solution of (143) is $\Lambda = \limsup_{t \to \infty} t^{-1} \log \|\epsilon_t\|$. The trivial solution $\epsilon_t = 0$ is *almost surely globally exponentially stable* if $\Lambda$ is almost surely negative for all $\epsilon_0 \in \mathbb{R}^{d_h}$.

For the sample Lyapunov exponent $\Lambda(\omega)$, there is a constant $C > 0$ and a random variable $0 \leq \tau(\omega) < \infty$ such that for all $t > \tau(\omega)$, $\|\epsilon_t\| = \|h_t' - h_t\| \leq Ce^{\Lambda t}$ almost surely. Therefore, almost sure exponential stability implies that almost all sample paths of (143) will tend to the equilibrium solution $\epsilon = 0$ exponentially fast. With this definition in tow, we state and prove our primary stability result, which is equivalent to Theorem 3 in the main paper.

**Theorem 13** (Bounds for sample Lyapunov exponent of the trivial solution). *Assume that Assumption C holds. Suppose that $a$ is $L_a$-Lipschitz, $0 \leq a_\Delta^T(\epsilon, t)\epsilon \leq L_a \|\epsilon\|_2^2$ and $0 \leq \sigma_1 \|\epsilon\| \leq \|\Delta \sigma_t(\epsilon)\|_F \leq \sigma_2 \|\epsilon\|$ for all nonzero $\epsilon \in \mathbb{R}^{d_h}$, $t \in [0, T]$. Then, with probability one,*

$$-\sigma_2^2 + \frac{\sigma_1^2}{2} + \lambda_{\min}(A^{\text{sym}}) \leq \Lambda \leq -\sigma_1^2 + \frac{\sigma_2^2}{2} + L_a \sigma_{\max}(W) + \lambda_{\max}(A^{\text{sym}}), \tag{145}$$

*for any $\epsilon_0 \in \mathbb{R}^{d_h}$.*

To establish the bounds in Theorem 13, we appeal to the following theorem, which arises from combining Theorems 4.3.3 and 4.3.5 in [23] in the case $p = 2$. Here, for a function $V$, we let $V_\epsilon = \partial V / \partial \epsilon$.

**Theorem 14** (Stochastic Lyapunov theorem). *If there exists a function $V \in \mathcal{C}^{2,1}(\mathbb{R}^{d_h} \times \mathbb{R}^+; \mathbb{R}^+)$ and $c_1, C_1 > 0$, $c_2, C_2 \in \mathbb{R}$, $c_3, C_3 \geq 0$ such that for all $\epsilon \neq 0$ and $t \geq t_0$,*

   *(i) $c_1 \|\epsilon\|^2 \leq V(\epsilon, t) \leq C_1 \|\epsilon\|^2$,*

   *(ii) $c_2 V(\epsilon, t) \leq LV(\epsilon, t) \leq C_2 V(\epsilon, t)$, and*

   *(iii) $c_3 V(\epsilon, t)^2 \leq \|V_\epsilon(\epsilon, t) \Delta \sigma_t(\epsilon)\|_F^2 \leq C_3 V(\epsilon, t)^2$,*

*then, with probability one, the Lyapunov exponent $\Lambda$ lies in the interval*

$$\frac{2c_2 - C_3}{4} \leq \Lambda \leq -\frac{c_3 - 2C_2}{4}. \tag{146}$$

The proof of Theorem 14 involves the Itô formula, an exponential martingale inequality and a Borel-Cantelli type argument. The functions $V$ above are called stochastic Lyapunov functions and the use of the theorem involves construction of these functions. We are now in a position to prove Theorem 13, and will find that the choice $V(\epsilon, t) = \|\epsilon\|^2$ will suffice.

*Proof of Theorem 13.* It suffices to verify the conditions of Theorem 2 with $V(\epsilon, t) = V(\epsilon) = \|\epsilon\|^2$.

Clearly (i) is satisfied. To show (iii), by the conditions on $\Delta\sigma_t$, we have that $4\sigma_1^2\|\epsilon\|^4 \leq \|V_\epsilon(\epsilon)\Delta\sigma_t(\epsilon)\|_F^2 \leq 4\sigma_2^2\|\epsilon\|^4$. It remains only to show (ii). Observe that

$$LV(\epsilon) = \epsilon^\top(A + A^\top)\epsilon + 2\Delta a_t(\epsilon)\epsilon + \mathrm{tr}(\Delta\sigma_t(\epsilon)\Delta\sigma_t(\epsilon)^\top).$$

Since $0 \leq \Delta a_t(\epsilon)\epsilon$ and

$$|\Delta a_t(\epsilon)\epsilon| \leq \|a(Wh_t' + Ux_t + b) - a(Wh_t + Ux_t + b)\|\,\|\epsilon\|$$
$$\leq L_a\,\|W\epsilon\|\,\|\epsilon\| \leq L_a\sigma_{\max}(W)\,\|\epsilon\|^2\,,$$

it follows that

$$LV(\epsilon) \leq (2\lambda_{\max}(A^{sym}) + 2L_a\sigma_{\max}(W) + \sigma_2^2)\|\epsilon\|^2,$$

and

$$LV(\epsilon) \geq (2\lambda_{\min}(A^{sym}) + \sigma_1^2)\|\epsilon\|^2.$$

The bound (146) now follows from Theorem 14 with $c_1 = C_1 = 1$, $c_2 = 2\lambda_{\min}(A^{sym}) + \sigma_1^2$, $C_2 = 2\lambda_{\max}(A^{sym}) + 2L_a\sigma_{\max}(W) + \sigma_2^2$, $c_3 = 4\sigma_1^2$, and $C_3 = 4\sigma_2^2$. $\qquad\square$

**Remark 5.** To see if the bounds in Theorem 13 are indeed sharp (at least for certain cases), consider the linear SDE $dH_t = AH_t dt + BH_t dW_t$, where $A \in \mathbb{R}^{d_h \times d_h}$, $B = \sigma I$, $\sigma \in \mathbb{R}$ and $W_t$ is a scalar Wiener process. Then, since $A$ and $B$ commute, they can be simultaneously diagonalized, and so the linear SDE can be reduced via transformation to a set of independent one-dimensional linear SDEs. In particular, one can show that $H_t = \exp((A - B^2/2)t + BW_t))H_0$ and the Lyapunov exponents $\Lambda$ of this system are the real part of the eigenvalues of $A - B^2/2$. Note that $\lambda_{\min}(A_{sym} - B^2/2) \leq \Lambda \leq \lambda_{\max}(A_{sym} - B^2/2)$ a.s.. Since $B = \sigma I$, this inequality implies Eqn. (145) with $L_a := 0$. The bounds are tight in the scalar case ($d_h = 1$ and $A$ is a scalar), with the inequality becoming an equality.

**Remark 6.** Even in the additive noise setting, however, the Lyapunov exponents of the CT-NRNN driven by additive noise are not generally the same as those of the corresponding deterministic CT-RNN. Oseledets multiplicative ergodic theorem implies they will be the same if the data generating process $x_t$ is ergodic [1]. Characterizing Lyapunov exponents for SDEs is a non-trivial affair in general — we refer to, for instance, [2] for details on this.

## G  Experimental Details

### G.1  Experimental Results Presented in the Main Paper

Following [6], we construct the hidden-to-hidden weight matrices $A$ and $W$ as

$$A = T(B, \beta_a, \gamma_a) := (1 - \beta_a) \cdot (B + B^T) + \beta_a \cdot (B - B^T) - \gamma_a I, \qquad (147)$$
$$W = T(C, \beta_w, \gamma_w) := (1 - \beta_w) \cdot (C + C^T) + \beta_w \cdot (C - C^T) - \gamma_w I. \qquad (148)$$

Here, $B$ and $C$ denote weight matrices that have the same dimensions as $A$ and $W$. The tuning parameters $\gamma_a$ and $\gamma_w$ can be used to increase dampening. We initialize the weight matrices by sampling weights from the normal distribution $\mathcal{N}(0, \sigma_{init}^2)$, where $\sigma_{init}^2$ is the variance. Table 4 summarizes the tuning parameters that we have used in our experiments. We train our models for 100 epochs, with scheduled learning rate decays at epochs $\{90\}$. We use Adam with default parameters for minimizing the objective.

We performed a random search to obtain the tuning parameters. Since our model is closely related to the Lipschitz RNN, we started with the tuning parameters proposed in [6]. We evaluated different noise levels, both for multiplicative and additive noise, in the range $[0.01, 0.1]$. We tuned the levels of noise-injection so that the models achieve state-of-the-art performance on clean input data. Further, we observed that the robustness of the model is not significantly improving when trained with

Table 4: Tuning parameters used to train the NRNN.

| Name | d_h | lr | decay | $\beta$ | $\gamma_a$ | $\gamma_w$ | $\epsilon$ | $\sigma^2_{init}$ | add. noise | mult. noise |
|------|-----|-----|-------|---------|-----------|-----------|-----------|-------------------|------------|-------------|
| Ordered MNIST | 128 | 0.001 | 0.1 | 0.75 | 0.001 | 0.001 | 0.01 | 0.1/128 | 0.02 | 0.02 |
| Ordered MNIST | 128 | 0.001 | 0.1 | 0.75 | 0.001 | 0.001 | 0.01 | 0.1/128 | 0.05 | 0.02 |
| Permuted MNIST | 128 | 0.001 | 0.1 | 0.75 | 0.001 | 0.001 | 0.01 | 0.1/128 | 0.02 | 0.02 |
| Permuted MNIST | 128 | 0.001 | 0.1 | 0.75 | 0.001 | 0.001 | 0.01 | 0.1/128 | 0.05 | 0.02 |
| ECG | 128 | 0.001 | 0.1 | 0.9 | 0.001 | 0.001 | 0.1 | 0.1/128 | 0.06 | 0.03 |

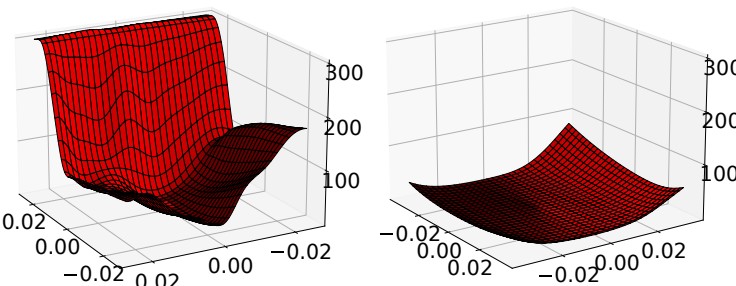

Figure 3: Hessian loss landscapes for deterministic (left) and noisy (right) model, computed using PyHessian.

increased levels of noise-injections. Overall, our experiments indicated that the model is relatively insensitive to the particular amount of additive and multiplicative noise level in the small noise regime.

Further, we need to note that we only considered models that used a combination of additive and multiplicative noise-injections. One could also train models using either only additive or multiplicative noise-injections. We did not investigate in detail the trade-offs between the different strategies. The motivation for our experiments was to demonstrate that (i) models trained with noise-injections can achieve state-of-the-art performance on clean input data, and (ii) such models are also more resilient to input perturbations.

Figure 3 shows that NRNN exhibits a smoother Hessian landscape than that of the deterministic counterpart.

For establishing a fair set of baselines, we used the following implementations and prescribed tuning parameters for the other models that we considered.

- **Exponential RNN.** We used the following implementation: `https://github.com/Lezcano/expRNN`. We used the default parameters. We trained the model, with hidden dimension $d_h = 128$, for 100 epochs.

- **CoRNN.** We used the following implementation, provided as part of the Supplementary Material: `https://openreview.net/forum?id=F3s69XzWOia`. We used the default parameters proposed by the authors for training the model with hidden dimension $d_h = 128$. We trained the model for 100 epochs with learning rate decay at epoch 90.

- **Lipschitz RNN.** We used the following implementation, provided as part of the Supplementary Material: `https://openreview.net/forum?id=-N7PBXqOUJZ`. We used the default parameters proposed by the authors for training the model with hidden dimension $d_h = 128$. We trained the model for 100 epochs with learning rate decay at epoch 90.

- **Antisymmetric RNN.** To our best knowledge, there is no public implementation by the authors for this model. However, the Antisymmetric RNN can be seen as a special case of the Lipschitz RNN or the NRNN, without the stabilizing term $A$ and without noise-injection. We trained this model by using our implementation and the following tuning parameters: $\beta = 1.0$, $\gamma = 0.001$, lr = 0.002, $\epsilon = 0.01$. We trained the model for 100 epochs with learning rate decay at epoch 90.

Table 5: Robustness w.r.t. white noise ($\sigma$) and S&P ($\alpha$) perturbations on the permuted MNIST task.

| Name | clean | $\sigma = 0.1$ | $\sigma = 0.2$ | $\sigma = 0.3$ | $\alpha = 0.03$ | $\alpha = 0.05$ | $\alpha = 0.1$ |
|---|---|---|---|---|---|---|---|
| Antisymmetric RNN [5] | 92.8% | 92.4% | 89.5% | 81.9% | 90.5% | 87.9% | 72.6% |
| CoRNN [24] | 96.05% | 65.1% | 38.25% | 29.1% | 84.8% | 73.8% | 52.6% |
| Exponential RNN [18] | 93.3% | 90.6% | 78.4% | 61.6% | 80.4% | 70.6% | 51.6% |
| Lipschitz RNN [6] | **95.9%** | 95.4% | 93.5% | 83.7% | 93.7% | 90.2% | 70.8% |
| NRNN (mult./add. noise: 0.02/0.02) | 94.9% | **94.8%** | **94.6%** | 94.3% | **94.0%** | 93.1% | 88.6% |
| NRNN (mult./add. noise: 0.02/0.05) | 94.7% | 94.6% | **94.6%** | **94.4%** | **94.0%** | **93.2%** | **90.5%** |

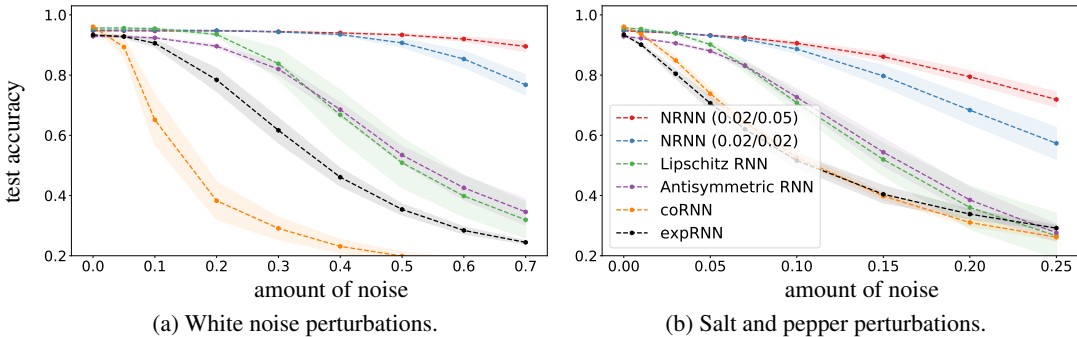

(a) White noise perturbations.  (b) Salt and pepper perturbations.

Figure 4: Test accuracy for the permuted MNIST task as function of the strength of input perturbations.

## G.2 Additional Results for Permuted Pixel-by-Pixel MNIST Classification

Here we consider the permuted pixel-by-pixel MNIST classification task. This task sequentially presents a scrambled sequence of the 784 pixels to the model and uses the final hidden state to predict the class membership probability of the input image.

Table 5 shows the average test accuracy (evaluated for models that are trained with 10 different seed values). Here we present results for white noise and salt and pepper (S&P) perturbations. Again, the NRNNs show an improved resilience to input perturbations. Figure 4 summarizes the performance of different models with respect to white noise and salt and pepper perturbations.