# OpenReview forum: "Noisy Recurrent Neural Networks"
_NeurIPS.cc/2021/Conference — NeurIPS 2021 Poster_

### Official Review · Reviewer_fjqb · 2021-07-13

**Rating:** 7
**Confidence:** 4

**Summary:**

This paper provides a theoretical framework for studying training of recurrent neural networks by injecting noise into hidden states.

**Ethical Concerns:**

None I could find.

**Limitations And Societal Impact:**

They didn't o it explicitely but I cannot see how this work can cause any negative societal impact.

**Main Review:**

The paper is very well written and organized. Some valuable technical results are provided to study discretization schemes of stochastic differential equations modeling the injection of noise into networks’ hidden states. These results justify the regularization effect of the injection scheme. This is an important result also illustrated by some representative simulations. They also clearly explain what the limitations are, such as the RELU activation function does not satisfy the conditions required in this paper.

**Time Spent Reviewing:**

1.5

---

> ### Author Response · Authors · 2021-08-10
> **Thank you for the kind feedback**
>
> We thank the reviewer very much for the positive feedback and appreciating our work, and for the favorable score. Indeed, ReLU does not satisfy our assumptions, but fortunately these are not typically used in the literature.

---

### Official Review · Reviewer_vC4r · 2021-07-14

**Rating:** 6
**Confidence:** 4

**Summary:**

This paper studies noise injection in RNNs, which can be regarded as a discretized SDE driven by external inputs. The first main result of the paper is the derivation of the implicit regularization caused by noise injection (similar to noise injection for feed-forward models), which is shown to be related to the Hessian of the loss (promoting flatter minima) and the Jacobian of the dynamics (promoting slower/more stable dynamics). A further implication on the classification margin is presented. Lastly, the paper presents a result showing that noise injection can further improve the temporal stability of the RNN dynamics. These motivate the use of the noise injection for recurrent neural networks, and experiments are provided to justify the theoretical results.

**Main Review:**

This paper is well written and the topics is of interest to the RNN/time series modelling community. Noise injection and its regularization effect is worthy of investigation, and in this sense, the main results of this paper presents advancements in this area. Below, I list some questions that the authors may wish to address:

1. It is well established in non-recurrent (or non-deep) models that noise injection leads to implicit regularization. Thus, it may be good to summarize the new phenomena arises from the dynamical setting you consider. One effect I can see from Theorem 2 is that the margin depends on the Jacobian of the dynamics - stable dynamics gives larger margin. I thought this is interesting. Are there any other differences between this dynamical setting and the usual setting in supervised learning?

2. Theorem 2: the result suggests that “stable” dynamics have bigger margins. There seems to be much more straightforward ways to achieve this, i.e. setting A in (3) to be very negative. Is the theorem saying something more than this?

3. Experiments

   1. Since the earlier results suggest that RNN with noise injection can be understood as implicitly regularization with the form of the regularizer as in (16), what if one simply adds a term such as (16) to the loss function. Does it achieve the same effect?

   1. Are there experiments to demonstrate the additional stabilization effect of noisy RNNs, i.e. Theorem 3?

---

After rebuttal: thank you for responding to my questions and my evaluation is unchanged.

**Time Spent Reviewing:**

3

---

> ### Author Response · Authors · 2021-08-10
> **Training on the edge of stability**
>
> We thank the reviewer for appreciating the presentation and recognizing the value of our paper, and for the favorable score.
>
> Regarding (1), we are glad that the reviewer finds the new phenomenon interesting and will make sure to highlight this in the revised version. We are not aware of any other differences between the dynamical setting and the usual setting in supervised learning, except that dependence of the Jacobian on the input data in the former setting (recurrent) differs from that in the latter setting (non-recurrent).
>
> Regarding (2), while stability is a desirable property according to Theorem 2, there is a tradeoff between stability and trainability in practice: the more stable the RNN is, the more pronounced the effects of vanishing gradient during training is, making capturing of long-term dependencies in input data more challenging. In particular, Emami et al. shows that training linear RNNs using gradient descent is implicitly biased towards short memory. One has to balance this stability-memory tradeoff in practice by working with models that are marginally stable (e.g., choosing A to be close to zero; how close depends on the task and dataset on hand). In fact, recent work such as the Antisymmetric RNN of Chang et al. and Lipschitz RNN of Erichson et al. focus on how to design RNN architectures to achieve a balance of this tradeoff.  We hope that our work can provide a framework to begin to answer those questions.
>
> Regarding (3.1), we believe that adding such terms to the loss function will achieve similar effects. Recent work of Camuto et al. in the setting of deep feedforward networks provides indications that this is indeed true. However, due to the complexity of the regularizer, assessing this in our case would be quite challenging.
>
> Regarding (3.2), we did not perform experiments to demonstrate the stochastic stabilization effects. This is due to the inherent difficulty of assessing these factors experimentally. The phenomenon is similar to the stabilizing effect of SGD in training neural networks, which is also quite difficult to observe. There is no additional incurred drift, so to observe the effect would require simulating the process on the edge of stability and showing stochastic boundedness of the trajectory of the hidden states. It’s unclear how to do this in a suitably convincing way. Hence, our experiments focus on the effect of noise-injections on robustness, which we believe is important but often neglected in the literature. It is also reasonable to demonstrate.

---

### Official Review · Reviewer_m9sN · 2021-07-14

**Rating:** 6
**Confidence:** 2

**Summary:**

This work studies RNNs that are trained with noise injection in the framework of ODEs.  The method is called NRNN. The authors provided Theorems to: (1) identify the form of implicit regularization of NRNN; (2) show the improved model stability; (3) provide the bounds for RNN classifiers. The experiments are conducted on two classification tasks: Sequential MNIST and heart beats classification.

**Main Review:**

The idea of viewing noise injection as stochastic learning is interesting. In general, this work is well presented, the main claims are well warranted by theoretical analysis and empirical results. In terms of the noise level in eq.(4), how to ensure the value \epsilon is small? How do the different combinations of the tunable parameters \sigma_1 and \sigma_2 influence the overall performance in Table 1-3?

The experimental results of robustness and resilience are interesting, NRNN presents better performance as compared to other methods while maintaining comparable predictive performance. I am wondering if this robustness can be held as well in the regression task?


**Time Spent Reviewing:**

2.5

---

> ### Author Response · Authors · 2021-08-10
> **The small noise regime provides a good tradeoff between accuracy and robustness**
>
> We thank the reviewer for appreciating the presentation and practical value of our paper, and for the favorable score.
>
> To clarify, the epsilon (controlling the noise level) is kept relatively small so that a perturbation analysis in the small noise regime is possible. In addition, choosing epsilon small enough has the benefit that the test performance on clean data is better or comparable to other state-of-the-art models. In the experiments, we explored noise levels in the range of 0.02 to 0.05 and found that a combination of additive and multiplicative noise seems to give rise to more robust models as compared to simply employing only additive or multiplicative noise. While we do not have a concrete theory for the optimal combination of noise, the intuition is that using both additive and multiplicative (which is also data-dependent) noise leads to patterns that have a rich structure. Importantly, our results demonstrate that this approach also helps to enhance robustness of the models to a wider family of data perturbations (e.g., salt and pepper noise or adversarial perturbations), i.e., data perturbations that have not been seen during training.
>
> Regarding your regression question, we believe that similar robustness results carry over to regression tasks in practice. However, we decided to focus on classification tasks in the present paper (providing both theory and experiments). Developing theory for the regression setting and a comprehensive empirical evaluation on regression tasks is an interesting direction for future work.

---

### Official Review · Reviewer_NYKx · 2021-07-17

**Rating:** 6
**Confidence:** 3

**Summary:**

The authors study the recurrent neural networks with injecting noise to hidden states. They consider RNNs as data driven discretization of stochastic differential equations. Analysis for global stability is presented, and it is also shown that noise injection may improve training stability.

**Limitations And Societal Impact:**

Yes

**Main Review:**

The analysis is based on the discretization of stochastic differential equations by adding the diffusion term on top of ordinary differential equations. With certain assumptions, the authors adopt this analysis to study the implicit regularization effect of noise injection schemes in the small noise regime, which promotes flatter minima, biases towards models with more stable dynamics, and favors larger classification margin in classification.

The approach of injecting noise in learning the highly non-convex problems can be an effective approach for obtaining more stable and generalizable models. The authors demonstrate a theoretical evidence for RNNs with injecting noise in states by applying the popular tools of differential equations, which marginally promotes our understanding in this regime. The empirical study also shows promising results of the proposed approach compared with existing ones, where under higher noise scenario the proposed approach significantly outperforms.

Though theoretically this is an interesting problem and remains with largely unresolved issues, the presented study does not reveal much new insights in the problem. The paper is relatively easy to follow and clearly presented. The presented proof is mostly clear and seems correct, though I did not check all detailed analysis for the proof.

**Time Spent Reviewing:**

5

---

> ### Author Response · Authors · 2021-08-10
> **Intriguing connection between the stability of RNNs and classification margin**
>
> We thank the reviewer for finding our paper interesting and easy to read, and for the favorable score.
>
> About whether our analysis reveals insight into the problem, two things.  First, we believe that our analysis of the effects of noise injection via the lens of implicit regularization does provide new insights into learning sequential data with Noisy RNNs. For example, as also pointed out by Reviewer vC4r, Theorem 2 reveals an intriguing connection between the stability of RNNs and classification margin. This is one of the most valuable insights (which is novel to the best of our knowledge) that we are trying to convey in the paper. We will better highlight this contribution in the revised version of the paper.
>
> Second, we not only provide theoretical analysis, but also supporting experimental results that demonstrate superior robustness of Noisy RNNs with respect to various types of data perturbations. We believe that it is important to study and improve the robustness of models rather than just focussing on improving the test accuracy on clean data.

---

> ### Comment · Reviewer_NYKx · 2021-08-26
> **I maintain my scores**
>
> Thanks for the discussion from the authors. After reading the responses, I tend to maintain scores, and still consider this is at the borderline.

---

> > ### Author Response · Authors · 2021-08-28
> > **Thank you for reading our response**
> >
> > Thank you very much for reading our response and reconsidering to change the score. We regret to hear that you still think that this paper is only slightly above the acceptance threshold, and we would like to better understand the reasons for this and what we can do to push the paper towards a clear accept.
> >
> > We developed a general framework for studying RNNs trained by injecting noise into hidden states. In turn we are able to make **three precise statements** about the implicit regularization effect of general noise injection schemes. In particular we find that **(1)** noise injection promotes flatter minima; **(2)** noise injections biases towards models with more stable dynamics; **(3)** in classification tasks, noise injections promote models with larger classification margin.
> >
> > While these insights might be intuitive to the reviewer, we are the first to make them precise. We strongly believe that the derived insights provide a significant **theoretical** contribution to the ML community. Indeed, you stated in your initial review that **”theoretically this is an interesting problem and remains with largely unresolved issues”**. Further, our theoretical insights are supported by experiments that show that noise injections are not only interesting in theory, but also lead to significantly more robust models in practice.

---

### Decision · Program_Chairs · 2021-09-27

**Decision:**

Accept (Poster)

**Comment:**

This paper received overall favorable reviews. This paper was found to be well written and the topic to be of interest and innovative.


Please address reviewers comment in the final version.